# Solving the Cold Start Problem on One's Own as an End User via Preference Transfer

**Ryoma Sato** *rsato@nii.ac.jp*
*National Institute of Informatics*

**Reviewed on OpenReview:** *https://openreview.net/forum?id=SgjOZdoVWH*

## Abstract

We propose a new approach that enables end users to directly solve the cold start problem by themselves. The cold start problem is a common issue in recommender systems, and many methods have been proposed to address the problem on the service provider's side. However, when the service provider does not take action, users are left with poor recommendations and no means to improve their experience. We propose an algorithm, PRETENDER, that allows end users to proactively solve the cold start problem on their own. PRETENDER does not require any special support from the service provider and can be deployed independently by users. We formulate the problem as minimizing the distance between the source and target distributions and optimize item selection from the target service accordingly. Furthermore, we establish theoretical guarantees for PRETENDER based on a discrete quadrature problem. We conduct experiments on real-world datasets to demonstrate the effectiveness of PRETENDER.

## 1 Introduction

Recommender systems have become an essential component of many online services, such as e-commerce [28, 31], social media [9, 59], and video streaming platforms [3, 15, 52]. These systems analyze user preferences and recommend items that the user may like. A common problem in recommender systems is the cold start problem, where the system cannot provide good recommendations for new users due to a lack of historical data. Many methods have been proposed to address the cold start problem, such as content-based filtering [24, 34, 49], utilizing side information [27, 37, 60], meta learning [25, 29, 53], and active learning [10, 61].

However, all of these methods are designed to be implemented on the service provider's side, and require the service provider to take action. When the service provider does not take action, e.g., due to lack of resources, lack of incentives, or simple negligence, users are left with poor recommendations and no immediate way to improve their experience. This can be particularly frustrating for users who are eager to adopt a new service but struggle to discover relevant items due to the cold start problem.

In this paper, we propose a new problem setting that enables users to address the cold start problem on their own. We consider a scenario where a user has been using a source service (e.g., Netflix) for a long time and has a history of preferences for items. The user has just started using a target service (e.g., Hulu) but lacks any preference history on that platform. The user receives good recommendations from the source service but poor recommendations from the target service due to the cold start problem. The user wants to transfer the preferences from the source service to the target service so that the user can enjoy good recommendations from the target service as well. If the target service offers a built-in functionality to import preferences, this problem can be easily solved. However, in many cases, such functionality is not provided, leaving the user with no direct means to improve their recommendations. To address this, we propose an algorithm, PRETENDER, that enables users to overcome the cold start problem independently, even when neither the source nor the target service provides dedicated support for preference transfer.

We formulate the problem as minimizing the distance between the source and target distributions and optimize the selection of items from the target service accordingly. A key strength of our method is that it provides strong theoretical guarantees even when the way the target service uses the data is unknown. We prove this by solving a new quadrature problem that arises in the optimization process, which is of independent interest. We conduct experiments on real-world datasets to demonstrate the effectiveness of PRETENDER. Our experiments show that PRETENDER can transfer the preferences effectively.

The contributions of this paper are as follows:

- We propose a new problem setting that enables users to independently address the cold start problem on their own.

- We propose PRETENDER, an algorithm that allows users to overcome the cold start problem with theoretical guarantees, even in the absence of support from the service provider.

- We conduct experiments on real-world datasets to validate the effectiveness of PRETENDER.

**Reproducibility**: Our code is available at `https://github.com/joisino/pretender`.

## 2 Problem Setting

Suppose we are an end user of the source service (e.g., Netflix) and the target service (e.g., Hulu). We have been using the source service for a long time and have a history of preferences for items (e.g., thumbs-up or thumbs-down for videos). We have just started using the target service and lack any preference history on that platform. We receive poor recommendations from the target service due to the cold start problem. Our goal is to transfer our preferences from the source service to the target service so that we can enjoy good recommendations from the target service as well. The problem is formally defined as follows:

**Problem (Preference Transfer).**
**Input**: Sets of source items $I_S$ and target items $I_T$. Features $\boldsymbol{x}_i$ of items $i \in I_S \cup I_T$. A set $\mathcal{D}_S = \{(i, y_i)\} \subset I_S \times \{0, 1\}$ of user preferences on source items. A positive integer $K \in \mathbb{Z}_+$ representing the number of target items to interact with.
**Output**: A set $\mathcal{D}_T = \{(i, y_i)\} \subset I_T \times \{0, 1\}$ with $|\mathcal{D}_T| = K$ such that clicking items following $\mathcal{D}_T$ results in good recommendations from the target service.

A method outputs a set of preferences for the target items, denoted as $\mathcal{D}_T = \{(i, y_i)\}$, Following this output, the end user clicks thumbs up for each item $i \in \{i \in I_T \mid (i, 1) \in \mathcal{D}_T\}$ and thumbs down for each item $i \in \{i \in I_T \mid (i, 0) \in \mathcal{D}_T\}$. A good method should output $\mathcal{D}_T$ such that the user can enjoy good recommendations from the target service after this process. Note that the clicking process (and thus the entire process) may be automated by a Web agent.

This problem presents three main challenges:

- **Items are not shared between services.** A naive approach might simply "copy" the history to the target service, but this approach fails because the corresponding items may not exist in the target service, i.e., $I_S \neq I_T$.

- **Clicking many items is tedious.** The user may not want to interact with many items just for preference transfer, i.e., $K$ is small. Even when the clicking process is automated, too large $K$ may take a long time and/or impose a heavy load on the target service.

- **The target service's use of the data is unknown.** The target service may process preference data $\mathcal{D}_T$ differently from the source service or in an unexpected manner, making it difficult to predict how the transferred preferences will influence recommendations.

# 3 Pretender

We propose PRETENDER (PREference Transfer by END usERs) to solve the preference transfer problem. We first describe the general framework and then provide variants for specific settings.

## 3.1 Formulating the Problem as Distance Minimization

The goal of PRETENDER is to select items from the target service such that its empirical distribution $\mu_T^{\mathcal{D}_T}$ is close to the source distribution $\mu_S$, which are defined as

$$\mu_T^{\mathcal{D}_T} = \frac{1}{K} \sum_{(i,y_i) \in \mathcal{D}_T} \delta_{(\boldsymbol{x}_i, y_i)}, \qquad \mu_S = \frac{1}{|\mathcal{D}_S|} \sum_{(i,y_i) \in \mathcal{D}_S} \delta_{(\boldsymbol{x}_i, y_i)} \tag{1}$$

where $\delta_x$ denotes the Dirac measure at $x$. We use the integral probability metric (IPM) [35, 51] to quantify the discrepancy between the distributions, which is defined for a function class $\mathcal{F}$ as

$$\text{IPM}_{\mathcal{F}}(\mu, \nu) = \sup_{f \in \mathcal{F}} \int f \, d\mu - \int f \, d\nu. \tag{2}$$

When $\mathcal{F}$ is the class of functions with a reproducing kernel hilbert space (RKHS) norm at most 1, the IPM is equivalent to the maximum mean discrepancy (MMD) [16], and when $\mathcal{F}$ is the class of 1-Lipschitz functions, the IPM corresponds to the 1-Wasserstein distance [39, 54]. PRETENDER selects items such that $\text{IPM}_{\mathcal{F}}(\mu_T, \mu_S)$ is small, ensuring that the target distribution closely aligns with the source distribution.

This formulation addresses the third challenge. Suppose the target service uses an unknown model $f_T(\cdot; \theta)$ and unknown loss function $\ell_T(\cdot, \cdot)$ to train the model. We only know that the loss $\ell_T(f_T(\boldsymbol{x}; \theta), y)$ is $L$-Lipschitz in $(\boldsymbol{x}, y)$. Then, the model incurs the following loss on the source preferences:

$$(\text{loss on the source data}) = \frac{1}{|\mathcal{D}_S|} \sum_{(i,y_i) \in \mathcal{D}_S} \ell_T(f_T(\boldsymbol{x}_i; \theta), y_i) \tag{3}$$

$$= \int \ell_T(f_T(\boldsymbol{x}; \theta), y) \, d\mu_S(\boldsymbol{x}, y) \tag{4}$$

$$\leq \int \ell_T(f_T(\boldsymbol{x}; \theta), y) \, d\mu_T^{\mathcal{D}_T}(\boldsymbol{x}, y) + L \cdot W_1(\mu_T^{\mathcal{D}_T}, \mu_S) \tag{5}$$

$$= (\text{training loss on the target data}) + L \cdot W_1(\mu_T^{\mathcal{D}_T}, \mu_S) \tag{6}$$

where $W_1$ denotes the 1-Wasserstein distance. The inequality follows from the definition of IPM (Eq. (2)). Therefore, if we minimize the Wasserstein distance $W_1(\mu_T, \mu_S)$ between the source and target distributions, and the target service effectively minimizes the training loss on the target data, the model trained on the target preferences will also generalize well to the source preferences, thereby accurately reflecting the user's preferences

The crux of this approach is that its guarantee is agnostic to the model, loss function, and the training method the target service employs, which are typically not known to the user. Regardless of how the target service uses the data, the user can ensure that the recommendation model trained on the target preferences reflects the source preferences as long as the distributions are close.

The use of IPM also addresses the first challenge. IPM exploits the geometry of the data space through the smoothness of the function class $\mathcal{F}$ (e.g., Lipschitz continuity and a small RKHS norm) and can be used even when the items are not shared between the services. This is in a stark contrast to other discrepancy measures such as KL divergence and Hellinger distance.

Although we have formulated the problem as distance minimization between distributions, this problem remains challenging because selection of items is combinatorial. This is in contrast to other minimization problems, where optimization is performed over weights and/or coordinates of data points, making the problem continuous and often convex. However, end users cannot "thumbs up 0.2 points" or alter the features of the items in the service, meaning that we cannot sidestep the combinatorial nature of the problem. We address this challenge in the following.

### 3.2 Optimization

The optimization framework of PRETENDER is as follows:

1. (**Item Preparation**) Prepare the set of labeled target items $J_T = \{(i, y) \mid i \in I_T, y \in \{0, 1\}\}$. We rearrange the items such that they are indexed by $1, 2, \ldots, 2m$ and redefine $J_T = \{(i_j, y_j) \mid j \in [2m]\}$, where $m = |I_T|$ is the number of items in the target service.

2. (**Continuous Optimization**) Optimize the weights $\boldsymbol{w} \in \left[0, \frac{1}{K}\right]^{2m} \cap \Delta_{2m}$ of the items in the target service such that the weighted empirical distribution $\mu_T^{\boldsymbol{w}} = \sum_{j=1}^{2m} \boldsymbol{w}_i \delta_{(\boldsymbol{x}_{i_j}, y_j)}$ is close to the source distribution $\mu_S$, where $\Delta_d$ denotes the $(d-1)$-dimensional probability simplex. This is achieved by solving the following optimization problem:

$$
\begin{aligned}
\min_{\boldsymbol{w} \in \mathbb{R}^{2m}} \; & \mathcal{D}\left(\sum_{j=1}^{2m} \boldsymbol{w}_i \delta_{(\boldsymbol{x}_{i_j}, y_j)}, \mu_S\right), \\
\text{s.t.} \quad & \sum_{j=1}^{2m} \boldsymbol{w}_j = 1, \qquad 0 \leq \boldsymbol{w}_j \leq \frac{1}{K} \; (j = 1, 2, \ldots, 2m).
\end{aligned}
\tag{7}
$$

3. (**Random Selection**) Sample items according to the optimized weights. For each $j \in [2m]$, sample $I_j \sim \text{Bernoulli}(K\boldsymbol{w}_j)$ for $j \in [2m]$, and define the selected set as $\hat{\mathcal{D}}_T = \{(i_j, y_j) \mid I_j = 1\}$.

4. (**Postprocessing**) If $|\hat{\mathcal{D}}_T| < K$, greedily insert additional items, and if $|\hat{\mathcal{D}}_T| > K$, greedily remove items to obtain the final preference set $\mathcal{D}_T$ satisfying $|\mathcal{D}_T| = K$.

**Item Preparation**. We construct the set of labeled target items as $J_T = \{(i_j, y_i) \mid j \in [2m]\}$. Since the user can choose the label for each item (e.g., thumbs up or thumbs down), we include both possible labels $(i, 0)$ and $(i, 1)$ in the candidate set, making the total number of items $2m$.

**Continuous Optimization**. We first solve the continuous relaxation of the problem. This problem is continuous and convex for MMD and the Wasserstein distance. Therefore, we can employ standard methods such as the Frank-Wolfe algorithm [22] or the projected subgradient descent algorithm [4] to solve the problem. By standard results in convex optimization, we can obtain $\hat{\boldsymbol{w}}$ such that

$$
\mathcal{D}(\mu_T^{\hat{\boldsymbol{w}}}, \mu_S) \leq \text{OPT}^{\text{continuous}} + \epsilon
\tag{8}
$$

$$
= \min_{\boldsymbol{w} \in \left[0, \frac{1}{K}\right]^{2m} \cap \Delta_{2m}} \mathcal{D}\left(\mu_T^{\boldsymbol{w}}, \mu_S\right) + \epsilon
\tag{9}
$$

$$
\overset{(a)}{\leq} \min_{\mathcal{D}_T = \{(i, y_i)\} : \, |\mathcal{D}_T| = K} \mathcal{D}\left(\frac{1}{K} \sum_{(i, y_i) \in \mathcal{D}_T} \delta_{(\boldsymbol{x}_i, y_i)}, \mu_S\right) + \epsilon
\tag{10}
$$

$$
= \text{OPT}^{\text{combinatorial}} + \epsilon,
\tag{11}
$$

where (a) follows because the weight $(1/K, \ldots, 1/K)$ of the empirical measure is in the feasible set. Therefore, the continuous solution is at least as good as the combinatorial solution. However, we need to carefully round the solution to obtain the final output to ensure that the rounding process does not degrade the quality of the solution much. This is the main challenge from a theoretical perspective.

**Random Selection**. We employ a randomized approach. We first point out that the distribution $\text{Bernoulli}(K\boldsymbol{w}_j)$ is well-defined as we set the optimization domain to $\boldsymbol{w} \in \left[0, \frac{1}{K}\right]^{2m} \cap \Delta_{2m}$ and $0 \leq K\boldsymbol{w}_j \leq 1$ holds. Let $\tilde{\boldsymbol{w}} \in \{0, 1/K\}^{2m}$ be the sample weights after the random selection, i.e., $\tilde{\boldsymbol{w}}_j = \frac{1}{K} I_j$. Then,

$$
\mathbb{E}[\tilde{\boldsymbol{w}}_j] = \frac{1}{K} \mathbb{E}[I_j] \overset{(a)}{=} \frac{1}{K} K \boldsymbol{w}_j = \boldsymbol{w}_j,
\tag{12}
$$

where (a) follows from $I_j \sim \text{Bernoulli}(K\boldsymbol{w}_j)$. Therefore, the expected weight is the same as the continuous solution. This implies that $\mathcal{D}(\mu_T^{\tilde{\boldsymbol{w}}}, \mu_S)$ distributes around that of the continuous solution $\mathcal{D}(\mu_T^{\hat{\boldsymbol{w}}}, \mu_S) \approx \text{OPT}^{\text{continuous}}$. Next, we analyze the number of selected items, which is given by, in expectation,

$$\mathbb{E}\left[\sum_{j=1}^{2m} I_j\right] = \sum_{j=1}^{2m} K\boldsymbol{w}_j \stackrel{\text{(a)}}{=} K, \tag{13}$$

where (a) follows from the fact that $\sum_{j=1}^{2m} \boldsymbol{w}_j = 1$, i.e., $\boldsymbol{w} \in \Delta_{2m}$. In addition,

$$\text{Var}\left[\sum_{j=1}^{2m} I_j\right] \stackrel{\text{(a)}}{=} \sum_{j=1}^{2m} \text{Var}[I_j] \stackrel{\text{(b)}}{=} \sum_{j=1}^{2m} K\boldsymbol{w}_j(1 - K\boldsymbol{w}_j) \stackrel{\text{(c)}}{\leq} \sum_{j=1}^{2m} K\boldsymbol{w}_j \stackrel{\text{(d)}}{=} K, \tag{14}$$

where (a) follows from the independence of the Bernoulli random variables, (b) follows from the variance of the Bernoulli random variables, (c) follows from $\boldsymbol{w}_j \geq 0$, and (d) follows from $\sum_{j=1}^{2m} \boldsymbol{w}_j = 1$. Therefore, the standard deviation is the order of $\sqrt{K}$, and the number of selected items is concentrated around $[K - O(\sqrt{K}), K + O(\sqrt{K})]$. This implies that the selection process does not blow up the error too much. We will elaborate this discussion in the following sections.

**Postprocessing**. Since the number of selected items may deviate from $K$, we may need to insert or remove items to make it exactly $K$. As the number of over- or under-selected items is on the order of $\sqrt{K}$, the error introduced by this step is also the order of $\frac{\sqrt{K}}{K} = K^{-1/2}$.

In the following sections, we will elaborate on the method and provide theoretical guarantees for MMD and the Wasserstein distance.

### 3.3 Pretender for MMD

We consider the case where we quantify the discrepancy between the distributions with MMD, defined as

$$\text{MMD}(\mu, \nu) = \sup_{\|f\|_{\mathcal{H}} \leq 1} \int f \, d\mu - \int f \, d\nu, \tag{15}$$

where $\mathcal{H}$ is the RKHS with the kernel $k$. Equivalently, the MMD can be expressed as

$$\text{MMD}(\mu, \nu)^2 = \left\| \int \phi(x) \, d\mu(x) - \int \phi(x) \, d\nu(x) \right\|_{\mathcal{H}}^2 \tag{16}$$

$$= \mathbb{E}_{x,x' \sim \mu}[k(x, x')] - 2\mathbb{E}_{x \sim \mu, x' \sim \nu}[k(x, x')] + \mathbb{E}_{x,x' \sim \nu}[k(x, x')], \tag{17}$$

where $\phi(x) = k(x, \cdot)$ is the feature map. Now, consider the empirical distributions

$$\mu^{\boldsymbol{w}} = \sum_{j=1}^{2m} \boldsymbol{w}_j \delta_{x_j}, \qquad \nu = \frac{1}{n} \sum_{j=1}^{n} \delta_{x'_j}, \tag{18}$$

where $x_j = (\boldsymbol{x}_{i_j}, y_j)$ is concatenation of the feature and label. Substituting these into Eq. (17), we obtain

$$\text{MMD}(\mu^{\boldsymbol{w}}, \nu)^2 = \sum_{j,j'=1}^{2m} \boldsymbol{w}_j \boldsymbol{w}_{j'} k(x_j, x_{j'}) - 2 \sum_{j=1}^{2m} \sum_{j'=1}^{n} \boldsymbol{w}_j \frac{1}{n} k(x_j, x'_{j'}) + \frac{1}{n^2} \sum_{j,j'=1}^{n} k(x'_j, x'_{j'}) \tag{19}$$

$$= \boldsymbol{w}^\top \boldsymbol{K}^{TT} \boldsymbol{w} - \frac{2}{n} \boldsymbol{1}^\top \boldsymbol{K}^{ST} \boldsymbol{w} + \frac{1}{n^2} \boldsymbol{1}^\top \boldsymbol{K}^{SS} \boldsymbol{1} \tag{20}$$

$$= \boldsymbol{w}^\top \boldsymbol{K}^{TT} \boldsymbol{w} - \frac{2}{n} \boldsymbol{1}^\top \boldsymbol{K}^{ST} \boldsymbol{w} + \text{const.}, \tag{21}$$

where $\boldsymbol{K}^{TT} \in \mathbb{R}^{2m \times 2m}$, $\boldsymbol{K}^{ST} \in \mathbb{R}^{2m \times n}$, and $\boldsymbol{K}^{SS} \in \mathbb{R}^{n \times n}$ are the kernel matrices with $\boldsymbol{K}^{TT}_{jj'} = k(x_j, x_{j'})$, $\boldsymbol{K}^{ST}_{jj'} = k(x_j, x'_{j'})$, and $\boldsymbol{K}^{SS}_{jj'} = k(x'_j, x'_{j'})$. These matrices are positive semi-definite, and $\mathrm{MMD}^2$ is convex in $\boldsymbol{w}$. Thus, the optimization problem to minimize MMD reduces to the following convex quadratic program:

$$\min_{\boldsymbol{w} \in \mathbb{R}^{2m}} \boldsymbol{w}^\top \boldsymbol{K}^{TT} \boldsymbol{w} - \frac{2}{n} \mathbf{1}^\top \boldsymbol{K}^{ST} \boldsymbol{w},$$

$$\text{s.t.} \quad \sum_{j=1}^{2m} \boldsymbol{w}_j = 1, \qquad 0 \le \boldsymbol{w}_j \le \frac{1}{K} \ (j = 1, 2, \ldots, 2m). \tag{22}$$

This problem can be solved with the Frank-Wolfe algorithm. We initialize $\boldsymbol{w} = \frac{1}{2m}$, which is feasible. We then iteratively update $\boldsymbol{w}$ by the Frank-Wolfe algorithm with step size $\frac{2}{t+2}$ for $t = 0, 1, \ldots, L - 1$. As the objective function is convex quadratic, we obtain the following guarantee.

**Proposition 3.1.** *When we run the Frank-Wolfe algorithm with the step size $\frac{2}{t+2}$ for $t = 0, 1, \ldots, L-1$, we obtain $\hat{\boldsymbol{w}}$ such that*

$$MMD(\mu_T^{\hat{\boldsymbol{w}}}, \mu_S) \le OPT^{continuous} + CL^{-1/2}, \tag{23}$$

*for some constant $C \in \mathbb{R}_+$.*

*Proof.* Since the objective function is quadratic, it is $\beta$-smooth. The Frank-Wolfe algorithm with a step size $\frac{2}{t+2}$ achieves a convergence rate of $O(L^{-1})$ [22]. Given that the objective function is the squared MMD, the error in MMD is on the order of $O(L^{-1/2})$. $\qquad\square$

As discussed in Section 3.2, we then round the solution by selecting items with probability $Kw_j$. To analyze this step, we make the following mild assumption.

**Assumption 3.2 (Bounded Kernel).** The kernel $k$ is bounded by $B$ in the sense that $k(x, x) = \langle \phi(x), \phi(x) \rangle = \|\phi(x)\|_{\mathcal{H}}^2 \le B$ for all $x \in \mathcal{X}$.

This assumption is mild and holds for many kernels such as the Gaussian, Laplacian, and Matérn kernels. We have the following guarantee for the random selection step.

**Proposition 3.3.** *Let $\tilde{\boldsymbol{w}}_j = \frac{I_j}{K}$ with $I_j \sim Bernoulli(K\hat{\boldsymbol{w}}_j)$. Then,*

$$\left\| \sum_{j=1}^{2m} \tilde{\boldsymbol{w}}_j \phi(x_j) - \sum_{j=1}^{2m} \hat{\boldsymbol{w}}_j \phi(x_j) \right\|_{\mathcal{H}} \le \sqrt{\frac{B}{\delta K}} \tag{24}$$

*with probability at least $1 - \delta$.*

*Proof Sketch.* We evaluate $\mathbb{E}_I \left[ \left\| \sum_j \tilde{\boldsymbol{w}}_j \phi(x_j) - \sum_j \hat{\boldsymbol{w}}_j \phi(x_j) \right\|_{\mathcal{H}}^2 \right]$. This can be bounded by $\frac{B}{K}$ by a similar argument to Eq. 14. Then we obtain the desired result by Markov's inequality. The full proof is provided in Appendix A. $\qquad\square$

We then analyze the postprocessing step. We have the following guarantee.

**Proposition 3.4.** *Let $\tilde{\boldsymbol{w}}'_j = \frac{1}{K}$ if the $j$-th item is in the final output $\mathcal{D}_T$ and $0$ otherwise. We have*

$$\left\| \sum_{j=1}^{2m} \tilde{\boldsymbol{w}}'_j \phi(x_j) - \sum_{j=1}^{2m} \tilde{\boldsymbol{w}}_j \phi(x_j) \right\|_{\mathcal{H}} \le \sqrt{\frac{B}{\delta K}} \tag{25}$$

*with probability at least $1 - \delta$.*

*Proof Sketch.* This also follows from a similar argument to Proposition 3.3. The proof is in Appendix B. $\qquad\square$

By combining the above guarantees, we have the following guarantee for the final output.

**Theorem 3.5.** *Under assumption 3.2, when we run the Frank-Wolfe algorithm with the step size $\frac{2}{t+2}$ for $t = 0, 1, \dots, L-1$ and select items with the probability $K\boldsymbol{w}_j$, we obtain $\mathcal{D}_T$ such that*

$$MMD(\mu_T^{\mathcal{D}_T}, \mu_S) \leq OPT^{combinatorial} + CL^{-1/2} + 2\sqrt{\frac{B}{\delta K}} \tag{26}$$

*with probability at least $1 - 2\delta$.*

*Proof Sketch.* This follows from the triangle inequality and union bound. The proof is in Appendix C. □

**Corollary 3.6.** *Under assumption 3.2, for any $\epsilon > 0$ and $K \in \mathbb{Z}_+$, there exists $L$ and $R$ such that by running the Frank-Wolfe algorithm with $L$ iterations and repeating the rounding process $R$ times and choose the best $\mathcal{D}_T^*$ with smallest $MMD(\mu_T^{\mathcal{D}_T^*}, \mu_S)$, we have*

$$MMD(\mu_T^{\mathcal{D}_T^*}, \mu_S) \leq OPT^{combinatorial} + 2\sqrt{\frac{2B}{K}} + \epsilon \tag{27}$$

$$= OPT^{combinatorial} + O(K^{-1/2}) \tag{28}$$

*with high probability.*

*Proof.* We set $L = 4C^2\varepsilon^{-2}$ and $\delta = \left(\frac{1}{\sqrt{2} + \frac{\varepsilon\sqrt{K}}{4\sqrt{B}}}\right)^2$, then we have

$$OPT^{combinatorial} + CL^{-1/2} + 2\sqrt{\frac{B}{\delta K}} \leq OPT^{combinatorial} + 2\sqrt{\frac{2B}{K}} + \epsilon \tag{29}$$

with probability at least $1 - 2\delta > 0$ because $\delta < \frac{1}{2}$. Repeating the rounding process $R = \Theta((1 - 2\delta)^{-1})$ times, we can obtain the final output with the desired guarantee. □

Note that the rounding process can be de-randomized by dynamic programming and the conditional probability method.

We can also bound $OPT^{combinatorial}$ under the following assumption.

**Assumption 3.7** (**Bounded Density Ratio**). The source items $\mathcal{D}_S$ and target candidate items $J_T$ are sampeled from distributions $P$ and $Q$, respectively, and the density ratio $r^* = \sup_x \frac{P(x)}{Q(x)}$ is bounded.

The items $\mathcal{D}_S$ and $J_T$ are not neccearily common (i.e., the first challenge), and the distributions are not necessarily identical. Instead, we assume that the target service has a chance to provide items similar to the source items with this assumption. Without this, it would be unlikely to find relevant items in the target service, making it impossible to transfer user preferences with a vanishing error. Under this assumption, we establish the following guarantee.

**Theorem 3.8.** *Under Assumptions 3.2 and 3.7, there exists $C_1, C_2 \in \mathbb{R}_+$ such that when $K \leq C_1 \frac{|J_T|}{r^*}$,*

$$MMD(P, \mu^{\mathcal{D}_T}) \leq C_2 \left( 2\sqrt{\frac{B}{|\mathcal{D}_S|}} + \sqrt{\frac{B}{K}} \right) \quad and \quad OPT^{combinatorial} \leq C_2 \left( \sqrt{\frac{B}{|\mathcal{D}_S|}} + \sqrt{\frac{B}{K}} \right) \tag{30}$$

*with high probability.*

*Proof Sketch.* Under Assumption 3.7, $\frac{1}{r^*}P \leq Q$ and $Q$ can be written as $\frac{1}{r^*}P + (1 - \frac{1}{r^*})Q'$ for some distribution $P'$. We essentially have $\frac{|J_T|}{r^*}$ samples from $P$ in the target service. We bound the convergence rate of the empirical measure to the true measure by a standard argument of MMD. The proof is in Appendix D. □

By combining the above guarantees, we have the following guarantee for the final output.

**Corollary 3.9.** *Under the assumptions of Corollary 3.6 and Theorem 3.8, we can obtain $\mathcal{D}_T$ such that*

$$MMD(\mu_T^{\mathcal{D}_T^*}, \mu_S) \leq O(|\mathcal{D}_S|^{-1/2} + K^{-1/2}) \tag{31}$$

*with high probability.*

Therefore, if we prepare a sufficient number of source items, the target service provides a sufficiently rich set of items, and we click sufficiently many items, we can transfer the preference to the target service with a vanishing error in the order of $O(|\mathcal{D}_S|^{-1/2} + K^{-1/2})$.

With the same argument as in Section 3.1, we can bound the error on the source items.

**Corollary 3.10.** *If the target service uses an unknown model $f_T(\cdot; \theta)$ and unknown loss function $\ell_T(\cdot, \cdot)$ to train the model, and the RKHS norm of loss $\ell_T(f_T(\boldsymbol{x}; \theta), y)$ is bounded by $R_\ell$, then the trained model incurs the error*

$$\mathbb{E}_P[\ell_T(f_T(\boldsymbol{x}; \theta), y)] \leq \frac{1}{|\mathcal{D}_T|} \sum_{(\boldsymbol{x}, y) \in \mathcal{D}_T} \ell_T(f_T(\boldsymbol{x}; \theta), y) + R_\ell \cdot MMD(P, \mathcal{D}_T). \tag{32}$$

*Under the assumptions of Theorem 3.8, we have*

$$\mathbb{E}_P[\ell_T(f_T(\boldsymbol{x}; \theta), y)] \leq \frac{1}{|\mathcal{D}_T|} \sum_{(\boldsymbol{x}, y) \in \mathcal{D}_T} \ell_T(f_T(\boldsymbol{x}; \theta), y) + O(|\mathcal{D}_S|^{-1/2} + K^{-1/2}). \tag{33}$$

This corollary shows that if the target service trains the model sufficiently well so that the training error $\frac{1}{|\mathcal{D}_T|} \sum_{(\boldsymbol{x}, y) \in \mathcal{D}_T} \ell_T(f_T(\boldsymbol{x}; \theta), y)$ is small and if we prepare a sufficient number of source items and click sufficiently many items following PRETENDER, we can make the right hand side of Eq. 33 arbitrarily small and we can ensure that the trained model incurs a small error on the source items, meaning that it effectively captures the users preferences from the source service.

### 3.4 Pretender for the Wasserstein Distance

We now consider the case where we quantify the discrepancy between the distributions with the 1-Wasserstein distance. Consider the empirical distributions $\mu_T^{\boldsymbol{w}}$ and $\mu_S$ defined in Eq. 18. The 1-Wasserstein distance is

$$W_1(\mu^{\boldsymbol{w}}, \nu) = \inf_{\boldsymbol{\gamma} \in \Pi(\mu^{\boldsymbol{w}}, \nu)} \sum_{j, j'} \boldsymbol{\gamma}_{jj'} \|x_j - x'_{j'}\| \tag{34}$$

where

$$\Pi(\mu^{\boldsymbol{w}}, \nu) = \left\{ \boldsymbol{\gamma} \in \mathbb{R}_{\geq 0}^{2m \times n} \,\middle|\, \boldsymbol{\gamma} \mathbf{1} = \boldsymbol{w}, \boldsymbol{\gamma}^\top \mathbf{1} = \frac{\mathbf{1}}{n} \right\} \tag{35}$$

is the set of coupling matrices. The 1-Wasserstein distance also admits the following dual formulation:

$$W_1(\mu^{\boldsymbol{w}}, \nu) = \sup_{f \in \mathcal{F}_{\text{Lip}}} \sum_{j=1}^{2m} \boldsymbol{w}_j f(x_j) - \frac{1}{n} \sum_{j=1}^{n} f(x'_j), \tag{36}$$

where $\mathcal{F}_{\text{Lip}}$ is the set of 1-Lipschitz functions. We solve the following convex optimization problem:

$$\begin{aligned} \min_{\boldsymbol{w} \in \mathbb{R}^{2m}} \quad & W_1(\mu^{\boldsymbol{w}}, \nu), \\ \text{s.t.} \quad & \sum_{j=1}^{2m} \boldsymbol{w}_j = 1, \qquad 0 \leq \boldsymbol{w}_j \leq \frac{1}{K} \ (j = 1, 2, \dots, 2m). \end{aligned} \tag{37}$$

By substituting the primal formulation (Eq. 34) and the definition of $\Pi(\mu^{\boldsymbol{w}}, \nu)$, we have

$$
\min_{\boldsymbol{w} \in \mathbb{R}^{2m}, \boldsymbol{\gamma} \in \mathbb{R}^{2m \times n}} \sum_{j,j'} \boldsymbol{\gamma}_{jj'} \|x_j - x'_{j'}\|,
$$

$$
\text{s.t.} \quad \boldsymbol{w}^\top \mathbf{1} = 1, \qquad \boldsymbol{\gamma}\mathbf{1} = \boldsymbol{w}, \qquad \boldsymbol{\gamma}^\top \mathbf{1} = \frac{\mathbf{1}}{n} \tag{38}
$$

$$
0 \le \boldsymbol{w}_j \le \frac{1}{K} \ (j = 1, 2, \ldots, 2m), \qquad \boldsymbol{\gamma}_{jj'} \ge 0 \ (j = 1, 2, \ldots, 2m, j' = 1, 2, \ldots, n).
$$

This problem is a linear program with optimization variables $\boldsymbol{w}$ and $\boldsymbol{\gamma}$. We can solve this problem by a linear program solver in a polynomial time. Let $\hat{\boldsymbol{w}}$ be the solution of this problem. We then select items with the probability $K\hat{\boldsymbol{w}}_j$ and analyze the rounding process. To analyze this step, we make the following mild assumption.

**Assumption 3.11** (Compact Domain). Each element $x_j$ lies is the unit cube $[0, 1]^d$ and the cost function $\|x - x'\|$ is the Euclidean distance.

If the data are not originally in the unit cube, they can be standardized to satisfy this assumption. To bound the Wasserstein distance, we need the following lemma, which is shown in Appendix E.

**Lemma 3.12.** *Let $\mathcal{F}_{Lip}$ be the set of 1-Lipschitz functions with $f(\frac{1}{2}) = 0$. $\mathcal{F}_{Lip}$ can be $L_\infty$-covered by $\tilde{\mathcal{F}}_\varepsilon$, i.e., for any $f \in \mathcal{F}_{Lip}$, there exists $\tilde{f} \in \tilde{\mathcal{F}}_\varepsilon$ such that $\|f - \tilde{f}\|_\infty \le \varepsilon$, with $|f(x)| \le \frac{\sqrt{d}}{2}, \forall f \in \tilde{\mathcal{F}}_\varepsilon$ and $|\tilde{\mathcal{F}}_\varepsilon| \le \exp\left(\log(3)\left(\frac{2\sqrt{d}}{\varepsilon}\right)^d\right)$.*

We first analyze the postprocessing step. We have the following guarantee.

**Proposition 3.13.** *Let $\tilde{\boldsymbol{w}}'_j = \frac{1}{K}$ if the $j$-th item is in the final output $\mathcal{D}_T$ and 0 otherwise. For any $\varepsilon > 0$,*

$$
\sup_{f \in \tilde{\mathcal{F}}_\varepsilon} \sum_{j=1}^{2m} f(x_j)(\tilde{\boldsymbol{w}}'_j - \tilde{\boldsymbol{w}}_j) \le \sqrt{\frac{d}{2K} \log\frac{1}{\delta}} + \frac{\sqrt{d}}{3K} \log\frac{1}{\delta} \tag{39}
$$

*with probability at least $1 - \delta$.*

*Proof Sketch.* The proof is similar to Proposition 3.4, and we obtain the bound with Bernstein's inequality. The full proof is in Appendix F. $\qquad\square$

We then analyze the random selection step. We first bound each test function.

**Lemma 3.14.** *Let $\tilde{\boldsymbol{w}}_j = \frac{I_j}{K}$ with $I_j \sim \text{Bernoulli}(K\hat{\boldsymbol{w}}_j)$. Then, for any $f$ with $|f(x)| \le \frac{\sqrt{d}}{2}$,*

$$
\sum_{j=1}^{2m} f(x_j)(\tilde{\boldsymbol{w}}_j - \hat{\boldsymbol{w}}_j) \le \sqrt{\frac{d}{2K} \log\frac{1}{\delta}} + \frac{\sqrt{d}}{3K} \log\frac{1}{\delta} \tag{40}
$$

*with probability at least $1 - \delta$.*

*Proof Sketch.* We bound the variance of independent random variables $f(x_j)(\tilde{\boldsymbol{w}}_j - \hat{\boldsymbol{w}}_j)$ $(j = 1, 2, \ldots, 2m)$ and apply Bernstein's inequality. The full proof is in Appendix G. $\qquad\square$

By combining Proposition 3.13, Lemma 3.14, and Lemma 3.12, we have the following guarantee.

**Theorem 3.15.** *For any $\delta > 0$, the 1-Wasserstein distance between the final output $\mathcal{D}_T$ and the source distribution is*

$$
W_1(\mu_T^{\mathcal{D}_T}, \mu_S) \le OPT^{combinatorial} + 3\sqrt{\frac{d}{K} \log\frac{1}{\delta}} + \frac{\sqrt{d}}{K} \log\frac{1}{\delta} + 8\sqrt{d}K^{-\frac{1}{d+2}} + 11\sqrt{d}K^{-\frac{2}{d+2}} + 6\sqrt{d \log\frac{1}{\delta}} K^{-\frac{1}{d+2}-\frac{1}{2}} \tag{41}
$$

$$
= OPT^{combinatorial} + O\left(K^{-\frac{1}{d+2}}\right) \tag{42}
$$

*with probability at least $1 - 2\delta$.*

*Proof Sketch.* We set $\varepsilon = \sqrt{d} \left( 2^{\frac{d-2}{d+2}} \log^{\frac{1}{d+2}}(3) K^{-\frac{1}{d+2}} + \frac{1}{2} \sqrt{\frac{1}{K} \log \frac{1}{\delta}} \right)$ in Lemma 3.12. We bound $|\tilde{\mathcal{F}}_\varepsilon|$, set $\delta \leftarrow \delta/|\tilde{\mathcal{F}}_\varepsilon|$ in Proposition 3.14, and apply the union bound. We then apply Proposition 3.13, Lemma 3.14, Lemma 3.12, and the Kantrovich-Rubinstein duality to obtain the desired result. The full proof is in Appendix H. $\qquad\square$

We can also bound the error of the trained model on the source items by the same argument as in Theorem 3.8 and Corollary 3.10. The difference is that the sample complexity of the Wasserstein distance is $\Theta(\frac{1}{\varepsilon^d})$ [12] instead of $\Theta(\frac{1}{\varepsilon^2})$ of the MMD. This is a fundamental property of the Wasserstein distance. Consequently, the optimal value $\mathrm{OPT}^{\mathrm{combinatorial}}$ and the loss bound scale as $\Theta(\frac{1}{\varepsilon^d})$ in the case of the Wasserstein distance, compared to $\Theta(\frac{1}{\varepsilon^2})$ for the MMD. This difference implies that the error bound can be significantly larger when the data dimension $d$ is high, making the Wasserstein distance less practical for high-dimensional settings.

## 4 Discussions

### 4.1 Discussion on Variants

We have presented the variants for MMD and the Wasserstein distance. Each has its own advantages and disadvantages. The primary advantage of MMD is its sample complexity. MMD is not affected by the number of dimensions of the feature space and sidesteps the curse of dimensionality while the Wasserstein distance suffers from it [12]. The advantage of the Wasserstein distance is its generality. The assumption that the loss function $\ell_T(f_T(\boldsymbol{x};\theta), y)$ is Lipschitz continuous is very mild and holds in many settings, such as logistic regression, matrix factorization, and neural networks. MMD requires the loss is in the RKHS, which may not hold in practice.

We also note that our analysis can be extended to other metrics as long as the continuous optimization problem (Eq. 7) is tractable and the covering number of the test function class $\mathcal{F}$ is bounded by the same argument of Section 3.4. For example, our analysis can be applied to the discrepancy distance [30], yielding a sharp generalization bound when the model is linear and the loss is the squared loss. Practitioners can choose metrics that best suits their specific application.

### 4.2 Optimum Value is Not Monotone

It should be noted that the optimum value of the combinatorial problem is not monotonic in $K$.

**Proposition 4.1.** *$OPT^{combinatorial}$ is not monotonic in $K$.*

*Proof.* We prove the proposition by a counterexample. Let $I_S = \{1\}, I_T = \{1, 2\}, \boldsymbol{x}_1 = 0, \boldsymbol{x}_2 = 1$, and $\mathcal{D}_S = \{(1, 1)\}$. When $K = 1$, the optimum value is 0 by selecting $\mathcal{D}_T = \{(1, 1)\}$. When $K = 2$, the optimum value is non-zero because we need to select items other than $(1, 1)$, and $\mathcal{D}_S \neq \mathcal{D}_T$. $\qquad\square$

One might intuitively expect that increasing $K$ would lead to a better or at least no worse solution. However, the above proposition shows that this is not always the case. In many practical situations, the goal is not to click exactly $K$ items, but rather to minimize the number of interactions while effectively transferring preferences. To address this issue, we can parallelly run the algorithm with $K' = 1, 2, \ldots, K$ and select the best solution with smallest $\mathcal{D}(\mu_T, \mu_S)$ among them.

### 4.3 Limitation: Inconsistent Features

We have assumed that we have access to the features $\boldsymbol{x}_i$ of the items $i$ in the problem setting. In practice, rich feature representations may not always be available, or the target service may utilize a different feature space than what we expect for training its recommendation model. This issue is particularly relevant when the target service employs collaborative filtering, a widely used approach in recommender systems. In such cases, the service may not use explicit features such as text, tags, or visual information, but use collaborative

features, which are not available to end users. In such a case, our approach and analysis cannot be directly applied. Nevertheless, we argue that our approach remains valuable even in such scenarios. First, explicit features often serve as good surrogates for implicit features. For example, movies with similar descriptions tend to attract similar audiences. Second, recent studies have demonstrated that even end users can estimate implicit features from recommendation networks [45]. These estimated features can then be fed into our approach. Extending our analysis to cases where feature spaces are inconsistent is an important direction for future work.

## 5 Related Work

### 5.1 Cold Start Problem

The cold start problem is a fundamental problem in recommender systems. It arises when a new user (or a new item) enters the system, and the system lacks sufficient information to provide accurate recommendations. Although many methods have been proposed to address the cold start problem [14, 26, 38, 55], all of them require the service providers to implement the method. In contrast, our approach is unique in that it can be applied directly by an end user without requiring any modifications to the service itself. This characteristic broadens the applicability of our method, allowing it to be used in services that lack built-in functionalities to address the cold start problem.

### 5.2 Quadrature

Quadrature is a technique to approximate the integral of a function by summing the function values at a finite number of points [1, 18, 19, 21, 41]. This is essentially equivalent to finding a discrete measure that approximates the given measure. Quadrature is widely used in numerical analysis and machine learning. One of the common applications is coresets [6, 23, 33, 50], which is a small set of training points that approximates the loss of the model on the entire training set. Our approach can be seen as a quadrature of $\int \ell_T \, d\mu_S$ with points $\mathcal{D}_T$. The main difference is that standard quadrature methods use arbitrary points and/or weights and sidestep combinatorial problems [2, 5], while our approach uses only the items in the target set, which naturally lead to the combinatorial optimization problem. For example, quadrature methods based on the Frank-Wolfe algorithm [2, 8, 58] output sparse weights, but they may choose the same items repeatedly in general, and the resulting weights are not uniform. Such output cannot be applied to our setting because users cannot thumbs up the same item multiple times in most services. Some approaches [13, 23] such as Kernel thinning [13] output a subset of the input points with uniform weights, which is similar to our approach. However, these methods assume that the candidate points are sampled according to the distribution being approximated. If this assumption does not hold, as in our case, these methods cannot be directly applied. Our proposed method can be used even when the input points are arbitrary. Other methods [56, 57] employ submodular optimization and greedy algorithms, achieving a $(1 - \frac{1}{e})$-approximation ratio. However, the gap of $(1 - \frac{1}{e})$ does not vanish as the number of items increases. By contrast, our approach can achieve the vanishing error as Corollary 3.10 shows thanks to the continuous optimization approach and the careful rounding process. To the best of our knowledge, our work is the first to provide a theoretical guarantee for such a general and combinatorial setting. We believe that this result is of independent interest.

### 5.3 User-side Realization

User-side realization refers to the concept in which end users implement desired functionalities on their own without requiring modifications to the service itself. Many users experience dissatisfaction with services. Even if they want some functionalities and request them to the service provider, the provider may not implement them due to various reasons such as cost, complexity, and simple negligence. After all, service providers are not volunteers but businesses. In such cases, the only options users have are not satisfactory, keep using the service despite their dissatisfaction or leave the service. User-side realization provides a proactive alternative to this dilemma. This concept has been explored in various fields such as recommender systems [43, 45, 47], search engines [11, 36, 40, 42, 44], and privacy [46]. The main advantage of the user-side realization is that it can be used in services that do not have special functionalities to address the problem,

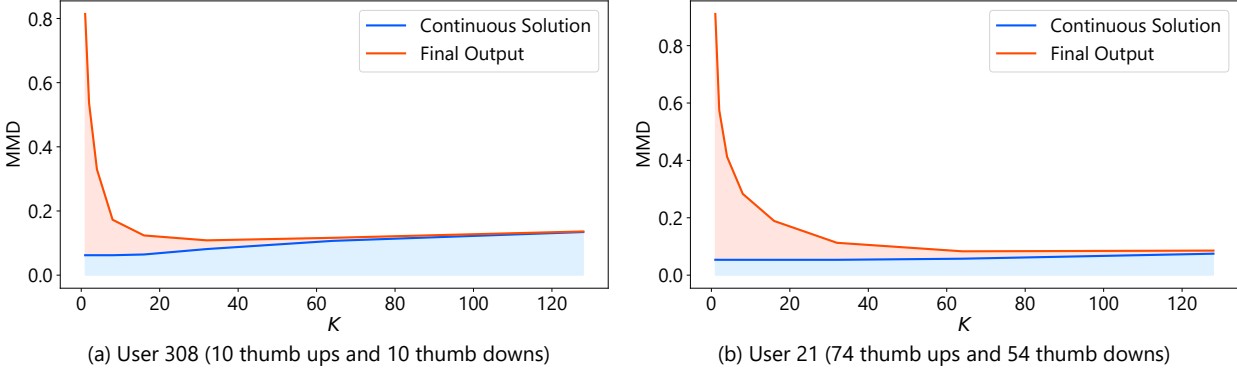

Figure 1: MMD as a function of the number of selected items $K$. The optimal value of the combinatorial optimization problem is intractable but is guaranteed to lie somewhere between the red and blue lines. We can see that the difference between the proposed method and the optimal value decreases as $K$ increases. As we discussed in Section 4.2, the optimal value is not monotonic in $K$, and the sweet spots lie around 32 to 64 in both cases. We can also see that the optimal values get worse more quickly in (a) because the source data contain fewer items, the relevant items in the target set are exhausted more quickly, and we are forced to select less relevant items when we increase $K$.

and it broadens the scope of the applicability of the solution. For a more comprehensive discussion on user-side realization, we refer the reader to the Ph.D. thesis by [48]. Our approach constitutes a novel application of user-side realization to the cold start problem in recommender systems.

## 6 Experiments

### 6.1 Convergence Analysis

We first confirm that the regret of PRETENDER converges to zero as the number $K$ of selected items increases.

**Experimental Setting.** We use the MovieLens 100K dataset [17]. We consider rating $\geq 4$ as positive feedback (i.e., thumbs up) and rating $< 4$ as negative feedback (i.e., thumbs down). We create two virtual services $S$ and $T$. We include each movie in $S$ with probability 0.5. We carry out the same process independently to create $T$. As a result, these services share about half of the movies. We use genres (e.g., Action, Adventure, Animation) and the release year as the features of the movies. They are encoded as a 90-dimensional multi-hot vector. We concatenate $Cy \in \{0, C\}$ to the feature vector to define the discrete distribution, where $y \in \{0, 1\}$ indicates thumbs up or thumbs down, and $C = 10$ is the hyperparameter that controls the emphasis on the label when we measure the distance between two data points. We focus on user 308, who is the first user who has 10 thumbs up and 10 thumbs down, and user 21, who is the first user who has more than 100 ratings. For each user $u$, we aim to transfer the preference $\mathcal{D}_u \cap \mathcal{D}_S$ by choosing $K$ items from $\mathcal{D}_T$. We use MMD with the Gaussian kernel and bandwidth $\sigma = 1$ as the distance measure. We use $L = 1000$ iterations of the Frank-Wolfe algorithm and $R = 100$ trials of the rounding process. We set $K = 1, 2, 4, 8, \ldots, 128$ to see the convergence behavior.

The goal of the experiment is to confirm that the regret of the proposed algorithm, i.e., the MMD of the proposed method minus the MMD of the optimal solution, converges to zero as the number of selected items $K$ increases. However, computing the true optimal solution is intractable due to the combinatorial nature of the optimization problem. Instead, we use the optimal value of the continuous relaxation as a benchmark, which serves as a provable lower bound for the optimal value of the combinatorial optimization problem.

**Results.** Figure 1 shows the regret of PRETENDER as a function of the number of selected items $K$. The optimal value of the combinatorial optimization problem is intractable but is guaranteed to lie somewhere between the red and blue lines. We observe that the difference between the proposed method and the optimal value decreases as $K$ increases. As we discussed in Section 4.2, the optimal value is not monotonic in $K$, and

Table 1. Performance Comparison. Each value represents the average MMD across all users. Lower is better. The standard deviation is computed across users. The proposed method is much better than the baseline methods and is close to the optimal continuous solution.

| | MovieLens with intersection | MovieLens no intersection | Last.fm no intersection | Amazon no intersection |
|---|---|---|---|---|
| Random | $0.2537 \pm 0.075$ | $0.2672 \pm 0.085$ | $0.3394 \pm 0.059$ | $0.1682 \pm 0.023$ |
| Greedy | $0.2978 \pm 0.074$ | $0.3401 \pm 0.086$ | $0.3555 \pm 0.098$ | $0.3043 \pm 0.069$ |
| Pretender | $\mathbf{0.1033 \pm 0.040}$ | $\mathbf{0.1250 \pm 0.045}$ | $\mathbf{0.0927 \pm 0.026}$ | $\mathbf{0.0774 \pm 0.010}$ |
| Continuous (unfeasible) | $0.0876 \pm 0.049$ | $0.1113 \pm 0.053$ | $0.0872 \pm 0.026$ | $0.0707 \pm 0.011$ |

Our proposed method, Pretender, performs much better than baselines.

Pretender gets close to the (unfeasible) lower bounds.
They provide provable certificates of Pretender's performance.

the sweet spots lie around 32 to 64 in both cases. We can also see that the optimal values get worse more quickly in (a) because the source preferences contain fewer items, and the relevant items in the target set are exhausted more quickly. Consequently, when $K$ increases, the algorithm is forced to select less relevant items in (a).

## 6.2 Quantitative Evaluation

**Datasets.** We evaluate our method using datasets from three domains: MovieLens 100k (movies), Last.fm (music) [7], and Amazon-Home-and-Kitchen (e-commerce) [20, 32]. We use the same setting as the convergence analysis for the MovieLens 100K dataset except that we use all users. For the Last.fm dataset, we consider musicians with positive weights as thumbs up and we sample the same number of random musicians for thumbs down. We use the tags (e.g., metal, 80's, chillout) as the features of the artists. As the number of tags is large, we apply the principal component analysis (PCA) to reduce the dimension to 50 and standardize the variance. For the Amazon-Home-and-Kitchen dataset, we consider ratings $\geq 4$ as thumbs up and ratings $< 4$ as thumbs down. We use the bag-of-words representation of the product reviews. We apply PCA to reduce the dimension to 50 and standardize the variance. We consider two different settings for defining source and target services. The first setting, with intersection, is identical to the one used in the convergence analysis. The second setting, no intersection, is that we split the items into source and target services without any intersection. This setting is more challenging because the methods cannot copy any source items to the target service. For Last.fm and Amazon-Home-and-Kitchen datasets, we use the Gaussian kernel with the bandwidth $\sigma = 10$ to reflect the scale of the features. We set $K = 100$ for all datasets.

**Baselines.** Since there are no existing methods that solve the cold start problem by an end user, we use two simple baselines to validate that the proposed method is reasonably effective. The first baseline is random selection, which randomly selects $K$ items from $\mathcal{D}_T$. The second baseline is the greedy selection, which selects $K$ items in $J_T$ that have the smallest distance $d(x, \mathcal{D}_S) = \min_{x' \in \mathcal{D}_S} \|x - x'\|_2$ to the source items $\mathcal{D}_S$. Additionally, we report the performance of the optimal solution of the continuous optimization problem.

**Results.** Table 1 shows MMD values between the source preference and the output of the methods. Lower is better. PRETENDER is much better than the baseline methods and is close to the optimal continuous solution. Note that checking the closeness is useful in practice as well. The continuous solution is obtained as a byproduct of PRETENDER. After we obtain the final output, we can verify that the final solution is indeed a reasonable one just by comparing the two values. We observe that the greedy method is worse than the random method. We hypothesize that it is because the greedy method chooses items that are too similar to some source items and the diversity is lost. We also observe that PRETENDER is robust to the setting with no intersection. The performance is slightly worse than in the setting with intersection but is still reasonablly well.

Table 2: **Case Study.** The source items the user interacted with and the $K = 20$ target items selected by PRETENDER. PRETENDER selects the very items that the user interacted with in the source service as shown in red. The user interacted with movies in 1996 and 1997, and PRETENDER mostly selects movies in 1996 and 1997. Notable exceptions are Alice in Wonderland (1951) and Fantasia (1940) selected by PRETENDER, as shown in blue. We hypothesize that this reflects the fact that the user thumbed down Cats Dont Dance (1997), which is an animated movie for children. Since Cats Don't Dance (1997) is not in the target service, PRETENDER selects similar movies from the target service and sends a signal that the user does not like this type of movies.

| Source Thumbs Up | Source Thumbs Down | Pretender Thumbs Up | Pretender Thumbs Down |
|---|---|---|---|
| Kolya (1996) | Ulee's Gold (1997) | Shall We Dance? (1996) | Absolute Power (1997) |
| The English Patient (1996) | Fly Away Home (1996) | The English Patient (1996) | Breakdown (1997) |
| G.I. Jane (1997) | Mrs. Brown (1997) | G.I. Jane (1997) | Fly Away Home (1996) |
| The Edge (1997) | Lost Highway (1997) | The Edge (1997) | Lost Highway (1997) |
| A Smile Like Yours (1997) | The Game (1997) | Das Boot (1981) | Alice in Wonderland (1951) |
| | Seven Years in Tibet (1997) | Addicted to Love (1997) | Fantasia (1940) |
| | Cats Don't Dance (1997) | The Pest (1997) | The Jackal (1997) |
| | Stag (1997) | | Seven Years in Tibet (1997) |
| | | | Paradise Road (1997) |
| | | | 'Til There Was You (1997) |
| | | | Le Bonheur (1965) |
| | | | All Over Me (1997) |
| | | | Rough Magic (1995) |

## 6.3 Case Study

As a case study, we use the MovieLens dataset and focus on user 308, who is the first user who has 10 thumbs up and 10 thumbs down. We investigate the target items selected by PRETENDER under the same experimental setup as Section 6.1 and with $K = 20$. Table 2 shows the source items the user interacted with and the target items selected by PRETENDER. Overall, PRETENDER suceeds in transferring the user's preferences. PRETENDER selects several items that the user previously interacted with in the source service, as highlighted in red. Note that the virtual services $S$ and $T$ share some items but not all items, and thus we cannot copy all items. We can also see that the user interacted with movies in 1996 and 1997, and PRETENDER mostly selects movies from the same period. This leads to similar preferences between the source and target services. Notable exceptions are Alice in Wonderland (1951) and Fantasia (1940) selected by PRETENDER, as shown in blue. We hypothesis that this reflects the fact that the user thumbed down Cats Don't Dance (1997), which is an animated movie for children. Since Cats Don't Dance (1997) is not in the target service, PRETENDER selects similar movies from the target service and sends a signal that the user does not like this type of movies to the terget service. This case study illustrates that PRETENDER effectively reflects the user's preferences.

# 7 Conclusion

In this paper, we made the following contributions.

- We proposed a new problem setting, solving the cold start problem on the user's side (Section 2).

- We proposed PRETENDER to solve the problem by formulating it as a metric minimization between distributions (Sections 3.1 − 3.2).

- We formally showed that the regret of PREDENTER converges to zero as the number of selected items increases (Sections 3.3 and 3.4).
  - We showed that the regret converges at a rate of $O(\frac{1}{\sqrt{K}})$ for MMD (Corollary 3.6) and $O(K^{-\frac{1}{d+2}})$ for the Wasserstein distance (Theorem 3.15).
  - We also showed that the optimal value also converges to zero as the number of source items and the number of selected items increases (Corollary 3.10).

- We empirically validated the effectiveness of PRETENDER through experiments (Sections 6.1 – 6.3).
  - We confirmed that the regret of the proposed algorithm converges to zero with a reasonable number of selected items (Section 6.1).
  - We showed that the proposed method is much better than baseline methods and is close to the optimal continuous solution with movie, music, and e-commerce datasets (Section 6.2).
  - We presented a case study, illustrating that the items selected by PRETENDER effectively reflect the user's preferences (Section 6.3).

We believe that this work introduces a new research direction for addressing the cold start problem from the users' perspective. We hope that our work will inspire further research in this area.

**Acknowledgements**

We thank Yuki Takezawa and Satoshi Hayakawa for helpful discussions.

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

# A    Proof of Proposition 3.3

**Proposition A.1.** *Let $\tilde{\boldsymbol{w}}_j = \frac{I_j}{K}$ with $I_j \sim Bernoulli(K\hat{\boldsymbol{w}}_j)$. Then,*

$$\left\| \sum_{j=1}^{2m} \tilde{\boldsymbol{w}}_j \phi(x_j) - \sum_{j=1}^{2m} \hat{\boldsymbol{w}}_j \phi(x_j) \right\|_{\mathcal{H}} \le \sqrt{\frac{B}{\delta K}} \tag{43}$$

*with probability at least $1 - \delta$.*

*Proof.*

$$\mathbb{E}_I \left[ \left\| \sum_j \tilde{\boldsymbol{w}}_j \phi(x_j) - \sum_j \hat{\boldsymbol{w}}_j \phi(x_j) \right\|_{\mathcal{H}}^2 \right] \tag{44}$$

$$= \sum_{j,j'} \mathbb{E}_I[\tilde{\boldsymbol{w}}_j \tilde{\boldsymbol{w}}_{j'}] \langle \phi(x_j), \phi(x_{j'}) \rangle - 2 \sum_{j,j'} \mathbb{E}_I[\tilde{\boldsymbol{w}}_j \hat{\boldsymbol{w}}_{j'}] \langle \phi(x_j), \phi(x_{j'}) \rangle + \sum_{j,j'} \mathbb{E}_I[\hat{\boldsymbol{w}}_j \hat{\boldsymbol{w}}_{j'}] \langle \phi(x_j), \phi(x_{j'}) \rangle \tag{45}$$

$$\overset{\text{(a)}}{=} \sum_{j,j'} \mathbb{E}_I \left[ \frac{I_j}{K} \frac{I_{j'}}{K} \right] \langle \phi(x_j), \phi(x_{j'}) \rangle - 2 \sum_{j,j'} \mathbb{E}_I \left[ \frac{I_j}{K} \hat{\boldsymbol{w}}_{j'} \right] \langle \phi(x_j), \phi(x_{j'}) \rangle + \sum_{j,j'} \mathbb{E}_I[\hat{\boldsymbol{w}}_j \hat{\boldsymbol{w}}_{j'}] \langle \phi(x_j), \phi(x_{j'}) \rangle \tag{46}$$

$$\overset{\text{(b)}}{=} \frac{1}{K^2} \sum_{j,j'} \mathbb{E}_I[I_j I_{j'}] \langle \phi(x_j), \phi(x_{j'}) \rangle - 2 \sum_{j,j'} \frac{1}{K} \mathbb{E}_I[I_j] \hat{\boldsymbol{w}}_{j'} \langle \phi(x_j), \phi(x_{j'}) \rangle + \sum_{j,j'} \hat{\boldsymbol{w}}_j \hat{\boldsymbol{w}}_{j'} \langle \phi(x_j), \phi(x_{j'}) \rangle \tag{47}$$

$$\overset{\text{(c)}}{=} \frac{1}{K^2} \sum_{j,j'} \mathbb{E}_I[I_j I_{j'}] \langle \phi(x_j), \phi(x_{j'}) \rangle - 2 \sum_{j,j'} \hat{\boldsymbol{w}}_j \hat{\boldsymbol{w}}_{j'} \langle \phi(x_j), \phi(x_{j'}) \rangle + \sum_{j,j'} \hat{\boldsymbol{w}}_j \hat{\boldsymbol{w}}_{j'} \langle \phi(x_j), \phi(x_{j'}) \rangle \tag{48}$$

$$= \frac{1}{K^2} \sum_{j,j'} \mathbb{E}_I[I_j I_{j'}] \langle \phi(x_j), \phi(x_{j'}) \rangle - \sum_{j,j'} \hat{\boldsymbol{w}}_j \hat{\boldsymbol{w}}_{j'} \langle \phi(x_j), \phi(x_{j'}) \rangle \tag{49}$$

$$= \frac{1}{K^2} \sum_j \mathbb{E}_I[I_j^2] \langle \phi(x_j), \phi(x_j) \rangle + \frac{1}{K^2} \sum_{j \neq j'} \mathbb{E}_I[I_j I_{j'}] \langle \phi(x_j), \phi(x_{j'}) \rangle - \sum_{j,j'} \hat{\boldsymbol{w}}_j \hat{\boldsymbol{w}}_{j'} \langle \phi(x_j), \phi(x_{j'}) \rangle \tag{50}$$

$$\stackrel{\text{(d)}}{=} \frac{1}{K^2} \sum_j \mathbb{E}_I[I_j] \langle \phi(x_j), \phi(x_j) \rangle + \frac{1}{K^2} \sum_{j \neq j'} \mathbb{E}_I[I_j] \mathbb{E}_I[I_{j'}] \langle \phi(x_j), \phi(x_{j'}) \rangle - \sum_{j,j'} \hat{\boldsymbol{w}}_j \hat{\boldsymbol{w}}_{j'} \langle \phi(x_j), \phi(x_{j'}) \rangle \tag{51}$$

$$\stackrel{\text{(e)}}{=} \frac{1}{K^2} \sum_j \mathbb{E}_I[I_j] \langle \phi(x_j), \phi(x_j) \rangle + \sum_{j \neq j'} \hat{\boldsymbol{w}}_j \hat{\boldsymbol{w}}_{j'} \langle \phi(x_j), \phi(x_{j'}) \rangle - \sum_{j,j'} \hat{\boldsymbol{w}}_j \hat{\boldsymbol{w}}_{j'} \langle \phi(x_j), \phi(x_{j'}) \rangle \tag{52}$$

$$= \frac{1}{K^2} \sum_j \mathbb{E}_I[I_j] \langle \phi(x_j), \phi(x_j) \rangle - \sum_j \hat{\boldsymbol{w}}_j^2 \langle \phi(x_j), \phi(x_j) \rangle \tag{53}$$

$$\stackrel{\text{(f)}}{=} \frac{1}{K} \sum_j \hat{\boldsymbol{w}}_j \langle \phi(x_j), \phi(x_j) \rangle - \sum_j \hat{\boldsymbol{w}}_j^2 \langle \phi(x_j), \phi(x_j) \rangle \tag{54}$$

$$= \frac{1}{K} \sum_j \hat{\boldsymbol{w}}_j \|\phi(x_j)\|_{\mathcal{H}}^2 - \sum_j \hat{\boldsymbol{w}}_j^2 \|\phi(x_j)\|_{\mathcal{H}}^2 \tag{55}$$

$$\stackrel{\text{(g)}}{\leq} \frac{1}{K} \sum_j \hat{\boldsymbol{w}}_j \|\phi(x_j)\|_{\mathcal{H}}^2 \tag{56}$$

$$\stackrel{\text{(h)}}{\leq} \frac{B}{K} \sum_j \hat{\boldsymbol{w}}_j \tag{57}$$

$$\stackrel{\text{(i)}}{=} \frac{B}{K}, \tag{58}$$

where (a) follows from the definition of $\tilde{\boldsymbol{w}}$, (b) follows from the fact that $\frac{1}{K}$ and $\hat{\boldsymbol{w}}$ are deterministic, (c) follows from $\mathbb{E}_I[I_j] = K\hat{\boldsymbol{w}}_j$, (d) follows from the fact that $I_j^2 = I_j$ and that $I_j$ and $I_{j'}$ are independent, (e) follows from $\mathbb{E}_I[I_j] = K\hat{\boldsymbol{w}}_j$, (f) follows from $\mathbb{E}_I[I_j] = K\hat{\boldsymbol{w}}_j$, (g) follows from $\hat{\boldsymbol{w}}_j^2 \|\phi(x_j)\|_{\mathcal{H}}^2 \geq 0$, (h) follows from Assumption 3.2. i.e., $\|\phi(x_j)\|_{\mathcal{H}}^2 \leq B$, and (i) follows from $\sum_j \hat{\boldsymbol{w}}_j = 1$. From Markov's inequality, we have

$$\mathbb{P}\left[\left\|\sum_{j=1}^{2m} \tilde{\boldsymbol{w}}_j \phi(x_j) - \sum_{j=1}^{2m} \hat{\boldsymbol{w}}_j \phi(x_j)\right\|_{\mathcal{H}} \geq \sqrt{\frac{B}{\delta K}}\right] \leq \delta. \tag{59}$$

$\square$

## B  Proof of Proposition 3.4

**Proposition B.1.** *Let $\tilde{\boldsymbol{w}}_j' = \frac{1}{K}$ if the $j$-th item is in the final output $\mathcal{D}_T$ and $0$ otherwise. We have*

$$\left\|\sum_{j=1}^{2m} \tilde{\boldsymbol{w}}_j' \phi(x_j) - \sum_{j=1}^{2m} \tilde{\boldsymbol{w}}_j \phi(x_j)\right\|_{\mathcal{H}} \leq \sqrt{\frac{B}{\delta K}} \tag{60}$$

*with probability at least $1 - \delta$.*

*Proof.*

$$\left\|\sum_{j=1}^{2m} \tilde{\boldsymbol{w}}_j' \phi(x_j) - \sum_{j=1}^{2m} \tilde{\boldsymbol{w}}_j \phi(x_j)\right\|_{\mathcal{H}} = \left\|\sum_{j=1}^{2m} \left(\tilde{\boldsymbol{w}}_j' - \tilde{\boldsymbol{w}}_j\right) \phi(x_j)\right\|_{\mathcal{H}} \tag{61}$$

$$\stackrel{\text{(a)}}{\leq} \sum_{j=1}^{2m} \left\|\left(\tilde{\boldsymbol{w}}_j' - \tilde{\boldsymbol{w}}_j\right) \phi(x_j)\right\|_{\mathcal{H}} \tag{62}$$

$$= \sum_{j=1}^{2m} |\tilde{\boldsymbol{w}}'_j - \tilde{\boldsymbol{w}}_j| \|\phi(x_j)\|_{\mathcal{H}} \tag{63}$$

$$\overset{(b)}{\le} \sqrt{B} \sum_{j=1}^{2m} |\tilde{\boldsymbol{w}}'_j - \tilde{\boldsymbol{w}}_j| \tag{64}$$

$$= \sqrt{B} \sum_{j=1}^{2m} \left| \frac{1}{K} 1[j \text{ is in the final output}] - \frac{1}{K} I_j \right| \tag{65}$$

$$= \frac{\sqrt{B}}{K} \sum_{j=1}^{2m} |1[j \text{ is in the final output}] - I_j| \tag{66}$$

$$\overset{(c)}{=} \frac{\sqrt{B}}{K} \left| K - \sum_{j=1}^{2m} I_j \right|, \tag{67}$$

where (a) follows from the triangle inequality, (b) follows from Assumption 3.2, and (c) follows from the fact that the non-zero elements of $\tilde{\boldsymbol{w}}'$ and $\tilde{\boldsymbol{w}}$ differ only when the corresponding items are included or removed in the postprocessing step, and the items are included or removed until the number of selected items is exactly $K$. From Eq. 13, $\mathbb{E}_I \left[ \sum_{j=1}^{2m} I_j \right] = K$, and from Eq. 14, $\mathrm{Var}_I \left[ \sum_{j=1}^{2m} I_j \right] \le K$. By Chebyshev's inequality to the right hand side of Eq. 67, we have

$$\mathbb{P} \left[ \left\| \sum_{j=1}^{2m} \tilde{\boldsymbol{w}}'_j \phi(x_j) - \sum_{j=1}^{2m} \tilde{\boldsymbol{w}}_j \phi(x_j) \right\|_{\mathcal{H}} \ge \sqrt{\frac{B}{\delta K}} \right] \le \delta. \tag{68}$$

$\square$

## C  Proof of Theorem 3.5

**Theorem C.1.** *Under assumption 3.2, when we run the Frank-Wolfe algorithm with the step size $\frac{2}{t+2}$ for $t = 0, 1, \ldots, L-1$ and select items with the probability $K\boldsymbol{w}_j$, we obtain $\mathcal{D}_T$ such that*

$$MMD(\mu_T^{\mathcal{D}_T}, \mu_S) \le OPT^{combinatorial} + CL^{-1/2} + 2\sqrt{\frac{B}{\delta K}} \tag{69}$$

*with probability at least $1 - 2\delta$.*

*Proof.*

$$\mathrm{MMD}(\mu_T^{\mathcal{D}_T}, \mu_S) \tag{70}$$

$$= \left\| \sum_{j=1}^{2m} \tilde{\boldsymbol{w}}'_j \phi(x_j) - \int \phi(x) \, d\mu_S(x) \right\|_{\mathcal{H}} \tag{71}$$

$$\overset{(a)}{\le} \left\| \sum_{j=1}^{2m} \tilde{\boldsymbol{w}}'_j \phi(x_j) - \sum_{j=1}^{2m} \tilde{\boldsymbol{w}}_j \phi(x_j) \right\|_{\mathcal{H}} + \left\| \sum_{j=1}^{2m} \tilde{\boldsymbol{w}}_j \phi(x_j) - \sum_{j=1}^{2m} \hat{\boldsymbol{w}}_j \phi(x_j) \right\|_{\mathcal{H}} + \left\| \sum_{j=1}^{2m} \hat{\boldsymbol{w}}_j \phi(x_j) - \int \phi(x) \, d\mu_S(x) \right\|_{\mathcal{H}} \tag{72}$$

$$= \left\| \sum_{j=1}^{2m} \tilde{\boldsymbol{w}}'_j \phi(x_j) - \sum_{j=1}^{2m} \tilde{\boldsymbol{w}}_j \phi(x_j) \right\|_{\mathcal{H}} + \left\| \sum_{j=1}^{2m} \tilde{\boldsymbol{w}}_j \phi(x_j) - \sum_{j=1}^{2m} \hat{\boldsymbol{w}}_j \phi(x_j) \right\|_{\mathcal{H}} + \mathrm{MMD}(\mu_T^{\hat{\boldsymbol{w}}}, \mu_S) \tag{73}$$

$$\overset{(b)}{\le} \left\| \sum_{j=1}^{2m} \tilde{\boldsymbol{w}}'_j \phi(x_j) - \sum_{j=1}^{2m} \tilde{\boldsymbol{w}}_j \phi(x_j) \right\|_{\mathcal{H}} + \left\| \sum_{j=1}^{2m} \tilde{\boldsymbol{w}}_j \phi(x_j) - \sum_{j=1}^{2m} \hat{\boldsymbol{w}}_j \phi(x_j) \right\|_{\mathcal{H}} + \mathrm{OPT}^{combinatorial} + CL^{-1/2} \tag{74}$$

$$\overset{(c)}{\leq} \sqrt{\frac{B}{\delta K}} + \sqrt{\frac{B}{\delta K}} + \text{OPT}^{\text{combinatorial}} + CL^{-1/2} \tag{75}$$

$$= \text{OPT}^{\text{combinatorial}} + CL^{-1/2} + 2\sqrt{\frac{B}{\delta K}}, \tag{76}$$

where (a) follows from the triangle inequality, (b) follows from Proposition 3.1, (c) follows from Propositions 3.3 and 3.4 and the union bound with probability at least $1 - 2\delta$. $\qquad\square$

## D  Proof of Theorem 3.8

**Theorem D.1.** *Under Assumptions 3.2 and 3.7, there exists $C_1, C_2 \in \mathbb{R}_+$ such that when $K \leq C_1 \frac{|J_T|}{r^*}$,*

$$MMD(P, \mu^{\mathcal{D}_T}) \leq C_2 \left( 2\sqrt{\frac{B}{|\mathcal{D}_S|}} + \sqrt{\frac{B}{K}} \right) \quad and \quad OPT^{combinatorial} \leq C_2 \left( \sqrt{\frac{B}{|\mathcal{D}_S|}} + \sqrt{\frac{B}{K}} \right) \tag{77}$$

*with high probability.*

*Proof.* Under Assumption 3.7, $\frac{1}{r^*}P \leq Q$ and $Q$ can be written as $\frac{1}{r^*}P + (1 - \frac{1}{r^*})Q'$ for some distribution $Q'$. Let $Z_i = 1$ with probability $\frac{1}{r^*}$ and $Z_i = 0$ with probability $1 - \frac{1}{r^*}$, and let sample $X_i$ from $P$ if $Z_i = 1$ and $Q'$ if $Z_i = 0$ for $i = 1, \ldots, |J_T|$. $X_i$ follows distribution $Q$. We identity $\{X_1, \ldots, X_{|J_T|}\}$ with $J_T$. Let $J_1 = \{X_i \mid Z_i = 1\}$. $\mathbb{E}\left[|J_1| = \frac{|J_T|}{r^*}\right]$ and $|J_1| \geq \frac{C_1 |J_T|}{r^*}$ for some constant $C_1$ with high probability by Markov's inequality. We assume this holds in the following. Suppose that $K \leq C_1 \frac{|J_T|}{r^*}$, then $K \leq |J_1|$. Let $J_2$ be the first $K$ samples from $J_1$. Elements in $J_2$ are independent and follow the distribution $P$. As the empirical distribution converges to the true distribution with rate $O(\sqrt{\frac{B}{K}})$ in MMD [16], we have

$$\text{MMD}(P, J_2) \leq C_2 \sqrt{\frac{B}{K}} \quad \text{and} \tag{78}$$

$$\text{MMD}(P, \mathcal{D}_S) \leq C_2 \sqrt{\frac{B}{|\mathcal{D}_S|}}, \tag{79}$$

for some constant $C_2$ with high probability. By the triangle inequality, we have

$$\text{MMD}(J_2, \mathcal{D}_S) \leq \text{MMD}(P, J_2) + \text{MMD}(P, \mathcal{D}_S) \tag{80}$$

$$\leq C_2 \left( \sqrt{\frac{B}{|\mathcal{D}_S|}} + \sqrt{\frac{B}{K}} \right). \tag{81}$$

As $J_2 \subseteq J_T$ and $|J_2| = K$, $J_2$ is a feasible solution of the combinatorial problem, and the optimal $\mathcal{D}_T \subseteq J_T$ has no worse MMD than $J_2$, and we have the desired guarantee. $\qquad\square$

## E  Proof of Lemma 3.12

**Lemma E.1.** *Let $\mathcal{F}_{Lip}$ be the set of 1-Lipschitz functions with $f(\frac{1}{2}) = 0$. $\mathcal{F}_{Lip}$ can be $L_\infty$-covered by $\tilde{\mathcal{F}}_\varepsilon$, i.e., for any $f \in \mathcal{F}_{Lip}$, there exists $\tilde{f} \in \tilde{\mathcal{F}}_\varepsilon$ such that $\|f - \tilde{f}\|_\infty \leq \varepsilon$, with $|f(x)| \leq \frac{\sqrt{d}}{2}, \forall f \in \tilde{\mathcal{F}}_\varepsilon$ and $|\tilde{\mathcal{F}}_\varepsilon| \leq \exp\left( \log(3) \left( \frac{2\sqrt{d}}{\varepsilon} \right)^d \right).$*

*Proof.* We devide the domain $[0,1]^d$ into $\lceil \frac{\sqrt{d}}{\varepsilon} \rceil^d$ hypercubes with diameter at most $\varepsilon$. For each hypercube $\mathcal{C}$ and $f \in \mathcal{F}_{\text{Lip}}$, let $f_{\mathcal{C}} = \frac{\max_{x \in \mathcal{C}} f(x) + \min_{x \in \mathcal{C}} f(x)}{2}$ and $\tilde{f}_{\mathcal{C}}$ be its nearest point in $\{k\varepsilon \mid k \in \mathbb{Z}\}$. We then define $\tilde{f}(x) = \tilde{f}_{\mathcal{C}}$ for $x \in \mathcal{C}$. We have $|f(x) - \tilde{f}(x)| \leq \varepsilon$ since the diameter of $\mathcal{C}$ is at most $\varepsilon$ and $f$ is 1-Lipschitz. We count the number of possible $\tilde{f}$. For the hypercube $\mathcal{C}_{\text{center}}$ with $\frac{1}{2} \in \mathcal{C}_{\text{center}}$, $\tilde{f}_{\mathcal{C}_{\text{center}}} = 0$ since $f(\frac{1}{2}) = 0$.

Given the value of $\tilde{f}_{\mathcal{C}}$, possible values of an adjacent hypercube $\mathcal{C}'$ are $\tilde{f}_{\mathcal{C}} - \varepsilon, \tilde{f}_{\mathcal{C}}, \tilde{f}_{\mathcal{C}} + \varepsilon$. We can reach all the hypercubes by repeating this process. Therefore, the possible values of $\tilde{f}$ are at most $3^{s-1}$, where $s = \lceil \frac{\sqrt{d}}{\varepsilon} \rceil^d$ is the number of hypercubes. When $\frac{\sqrt{d}}{\varepsilon} \geq 1$, we have $\lceil \frac{\sqrt{d}}{\varepsilon} \rceil \leq \frac{\sqrt{d}}{\varepsilon} + 1 \leq \frac{2\sqrt{d}}{\varepsilon}$, and $|\tilde{\mathcal{F}}| \leq \exp\left( \log(3) \left( \frac{2\sqrt{d}}{\varepsilon} \right)^d \right)$. When $\frac{\sqrt{d}}{\varepsilon} < 1$, we have $|\tilde{\mathcal{F}}| = 1$, in which case the lemma holds trivially. $\square$

## F   Proof of Proposition 3.13

**Proposition F.1.** *Let $\tilde{\boldsymbol{w}}'_j = \frac{1}{K}$ if the $j$-th item is in the final output $\mathcal{D}_T$ and $0$ otherwise. For any $\varepsilon > 0$,*

$$\sup_{f \in \tilde{\mathcal{F}}_\varepsilon} \sum_{j=1}^{2m} f(x_j)(\tilde{\boldsymbol{w}}'_j - \tilde{\boldsymbol{w}}_j) \leq \sqrt{\frac{d}{2K} \log \frac{1}{\delta}} + \frac{\sqrt{d}}{3K} \log \frac{1}{\delta} \tag{82}$$

*with probability at least $1 - \delta$.*

*Proof.*

$$\sup_{f \in \tilde{\mathcal{F}}_\varepsilon} \sum_{j=1}^{2m} (\tilde{\boldsymbol{w}}'_j - \tilde{\boldsymbol{w}}_j) f(x_j) \leq \|f\|_\infty \sum_{j=1}^{2m} |\tilde{\boldsymbol{w}}'_j - \tilde{\boldsymbol{w}}_j| \tag{83}$$

$$\leq \frac{\sqrt{d}}{2} \sum_{j=1}^{2m} |\tilde{\boldsymbol{w}}'_j - \tilde{\boldsymbol{w}}_j| \tag{84}$$

$$\leq \frac{\sqrt{d}}{2} \sum_{j=1}^{2m} \left| \frac{1}{K} 1[j \text{ is in the final output}] - \frac{1}{K} I_j \right| \tag{85}$$

$$= \frac{\sqrt{d}}{2K} \sum_{j=1}^{2m} |1[j \text{ is in the final output}] - I_j| \tag{86}$$

$$= \frac{\sqrt{d}}{2K} \left| K - \sum_{j=1}^{2m} I_j \right|. \tag{87}$$

$I_j$ are independent Bernoulli random variables with $|I_j| \leq 1$, and from Eq. 13, $\mathbb{E}_I \left[ \sum_{j=1}^{2m} I_j \right] = K$, and from Eq. 14, $\mathrm{Var}_I \left[ \sum_{j=1}^{2m} I_j \right] \leq K$. By Bernstein's inequality, we have

$$\mathbb{P}\left[ \left| K - \sum_{j=1}^{2m} I_j \right| \geq \sqrt{2K \log \frac{1}{\delta}} + \frac{2}{3} \log \frac{1}{\delta} \right] \leq \delta. \tag{88}$$

By combining it with Eq. 87, we have the desired guarantee. $\square$

## G   Proof of Proposition 3.14

**Lemma G.1.** *Let $\tilde{\boldsymbol{w}}_j = \frac{I_j}{K}$ with $I_j \sim Bernoulli(K\hat{\boldsymbol{w}}_j)$. Then, for any $f$ with $|f(x)| \leq \frac{\sqrt{d}}{2}$,*

$$\sum_{j=1}^{2m} f(x_j)(\tilde{\boldsymbol{w}}_j - \hat{\boldsymbol{w}}_j) \leq \sqrt{\frac{d}{2K} \log \frac{1}{\delta}} + \frac{\sqrt{d}}{3K} \log \frac{1}{\delta} \tag{89}$$

*with probability at least $1 - \delta$.*

*Proof.* Since $0 \leq \tilde{\boldsymbol{w}}_j, \hat{\boldsymbol{w}}_j \leq \frac{1}{K}$, we have $|(\tilde{\boldsymbol{w}}_j - \hat{\boldsymbol{w}}_j)f(x_j)| \leq \frac{\sqrt{d}}{2K}$. Here, $(\tilde{\boldsymbol{w}}_j - \hat{\boldsymbol{w}}_j)f(x_j)$ are independent random variables with mean $\mathbb{E}_I[(\tilde{\boldsymbol{w}}_j - \hat{\boldsymbol{w}}_j)]f(x_j) = 0$ and variance

$$\text{Var}_I[(\tilde{\boldsymbol{w}}_j - \hat{\boldsymbol{w}}_j)f(x_j)] = f(x_j)^2 \text{Var}_I[\tilde{\boldsymbol{w}}_j] = f(x_j)^2 \text{Var}_I\left[\frac{I_j}{K}\right] = \frac{f(x_j)^2}{K^2}\text{Var}_I[I_j] = \frac{f(x_j)^2 K \hat{\boldsymbol{w}}_j(1 - K\hat{\boldsymbol{w}}_j)}{K^2}. \tag{90}$$

The sum of the variances is

$$\sum_{j=1}^{2m} \frac{f(x_j)^2 K \hat{\boldsymbol{w}}_j(1 - K\hat{\boldsymbol{w}}_j)}{K^2} = \frac{1}{K}\sum_{j=1}^{2m} f(x_j)^2 \hat{\boldsymbol{w}}_j(1 - K\hat{\boldsymbol{w}}_j) \tag{91}$$

$$\overset{(a)}{\leq} \frac{d}{4K}\sum_{j=1}^{2m} \hat{\boldsymbol{w}}_j(1 - K\hat{\boldsymbol{w}}_j) \tag{92}$$

$$\overset{(b)}{\leq} \frac{d}{4K}\sum_{j=1}^{2m} \hat{\boldsymbol{w}}_j \tag{93}$$

$$\overset{(c)}{=} \frac{d}{4K}, \tag{94}$$

where (a) follows from $f(x_j) \leq \frac{\sqrt{d}}{2}$, (b) follows from the fact that $1 \geq K\hat{\boldsymbol{w}}_j$, and (c) follows from $\sum_j \hat{\boldsymbol{w}}_j = 1$. By Bernstein's inequality, we have

$$\mathbb{P}\left[\sum_{j=1}^{2m}(\tilde{\boldsymbol{w}}_j - \hat{\boldsymbol{w}}_j)f(x_j) \geq \sqrt{\frac{d}{2K}\log\frac{1}{\delta}} + \frac{\sqrt{d}}{3K}\log\frac{1}{\delta}\right] \leq \delta. \tag{95}$$

$\square$

## H Proof of Theorem 3.15

**Theorem H.1.** *For any $\delta > 0$, the 1-Wasserstein distance between the final output $\mathcal{D}_T$ and the source distribution is*

$$W_1(\mu_T^{\mathcal{D}_T}, \mu_S) \leq OPT^{combinatorial} + 3\sqrt{\frac{d}{K}\log\frac{1}{\delta}} + \frac{\sqrt{d}}{K}\log\frac{1}{\delta} + 8\sqrt{d}K^{-\frac{1}{d+2}} + 11\sqrt{d}K^{-\frac{2}{d+2}} + 6\sqrt{d\log\frac{1}{\delta}}K^{-\frac{1}{d+2}-\frac{1}{2}} \tag{96}$$

*with probability at least $1 - 2\delta$.*

*Proof.* We use

$$\varepsilon = \sqrt{d}\left(2^{\frac{d-2}{d+2}}\log^{\frac{1}{d+2}}(3)K^{-\frac{1}{d+2}} + \frac{1}{2}\sqrt{\frac{1}{K}\log\frac{1}{\delta}}\right) \tag{97}$$

in the following. Then,

$$\frac{4K}{d}\varepsilon^{d+2} \geq 4Kd^{-1}d^{\frac{d+2}{2}}\left(2^{\frac{d-2}{d+2}}\log^{\frac{1}{d+2}}(3)K^{-\frac{1}{d+2}}\right)^{d+2} \tag{98}$$

$$= d^{\frac{d}{2}}2^d\log(3) \tag{99}$$

and thus

$$\left(\frac{2\sqrt{d}}{\varepsilon}\right)^d\log(3) \leq \frac{4K}{d}\varepsilon^2. \tag{100}$$

In addition,

$$\frac{4K}{d}\varepsilon^2 \ge 4K \left( \frac{1}{2} \sqrt{\frac{1}{K} \log \frac{1}{\delta}} \right)^2 \tag{101}$$

$$= \log \frac{1}{\delta}. \tag{102}$$

By combining Eq. 99 and Eq. 102, we have

$$\left( \frac{2\sqrt{d}}{\varepsilon} \right)^d \log(3) + \log \frac{1}{\delta} \le \frac{8K}{d}\varepsilon^2. \tag{103}$$

By Lemma 3.12, $|\tilde{\mathcal{F}}_\varepsilon| \le \exp\left( \log(3) \left( \frac{2\sqrt{d}}{\varepsilon} \right)^d \right)$. We have

$$\frac{d}{2K} \log \frac{|\tilde{\mathcal{F}}_\varepsilon|}{\delta} = \frac{d}{2K} (\log |\tilde{\mathcal{F}}_\varepsilon| + \log \frac{1}{\delta}) \tag{104}$$

$$\le \frac{d}{2K} \left( \log(3) \left( \frac{2\sqrt{d}}{\varepsilon} \right)^d + \log \frac{1}{\delta} \right) \tag{105}$$

$$\overset{(a)}{\le} \frac{d}{2K} \left( \frac{8K}{d}\varepsilon^2 \right) \tag{106}$$

$$= 4\varepsilon^2, \tag{107}$$

where (a) follows from Eq. 103. In addition, we have

$$\frac{\sqrt{d}}{3K} \log \frac{|\tilde{\mathcal{F}}_\varepsilon|}{\delta} = \frac{\sqrt{d}}{3K} (\log |\tilde{\mathcal{F}}_\varepsilon| + \log \frac{1}{\delta}) \tag{108}$$

$$\le \frac{\sqrt{d}}{3K} \left( \log(3) \left( \frac{2\sqrt{d}}{\varepsilon} \right)^d + \log \frac{1}{\delta} \right) \tag{109}$$

$$\overset{(a)}{\le} \frac{\sqrt{d}}{3K} \left( \frac{8K}{d}\varepsilon^2 \right) \tag{110}$$

$$= \frac{8}{3\sqrt{d}}\varepsilon^2, \tag{111}$$

where (a) follows from Eq. 103. We set $\delta \leftarrow \delta/|\tilde{\mathcal{F}}|$ in Proposition 3.14 and apply the union bound, then, we have

$$\sum_{j=1}^{2m} (\tilde{\boldsymbol{w}}_j - \hat{\boldsymbol{w}}_j) \tilde{f}(x_j) \le \sqrt{\frac{d}{2K} \log \frac{|\tilde{\mathcal{F}}|}{\delta}} + \frac{\sqrt{d}}{3K} \log \frac{|\tilde{\mathcal{F}}|}{\delta} \tag{112}$$

$$\overset{(a)}{\le} 2\varepsilon + \frac{8}{3\sqrt{d}}\varepsilon^2 \tag{113}$$

for all $\tilde{f} \in \tilde{\mathcal{F}}$ with probability at least $1 - \delta$, where (a) follows from Eq. 107 and Eq. 111. In the following, we assume that Eqs. 39 and 113 hold, which is true with probability at least $1 - 2\delta$. Then, we have

$$\sum_{j=1}^{2m} (\tilde{\boldsymbol{w}}'_j - \hat{\boldsymbol{w}}_j) \tilde{f}(x_j) = \sum_{j=1}^{2m} (\tilde{\boldsymbol{w}}'_j - \tilde{\boldsymbol{w}}_j) \tilde{f}(x_j) + \sum_{j=1}^{2m} (\tilde{\boldsymbol{w}}_j - \hat{\boldsymbol{w}}_j) \tilde{f}(x_j) \tag{114}$$

$$\le \sqrt{\frac{d}{2K} \log \frac{1}{\delta}} + \frac{\sqrt{d}}{3K} \log \frac{1}{\delta} + 2\varepsilon + \frac{8}{3\sqrt{d}}\varepsilon^2. \tag{115}$$

For any $f \in \mathcal{F}_{\text{Lip}}$, there exists $\tilde{f} \in \tilde{\mathcal{F}}$ such that $\|f - \tilde{f}\|_\infty \le \varepsilon$, and we have

$$\sum_{j=1}^{2m} (\tilde{\boldsymbol{w}}'_j - \hat{\boldsymbol{w}}_j) f(x_j) \tag{116}$$

$$= \sum_{j=1}^{2m} (\tilde{\boldsymbol{w}}'_j - \hat{\boldsymbol{w}}_j) \tilde{f}(x_j) + \sum_{j=1}^{2m} (\tilde{\boldsymbol{w}}'_j - \hat{\boldsymbol{w}}_j)(f(x_j) - \tilde{f}(x_j)) \tag{117}$$

$$\overset{(a)}{\le} \sum_{j=1}^{2m} (\tilde{\boldsymbol{w}}'_j - \hat{\boldsymbol{w}}_j) \tilde{f}(x_j) + \varepsilon \sum_{j=1}^{2m} |\tilde{\boldsymbol{w}}'_j - \hat{\boldsymbol{w}}_j| \tag{118}$$

$$\overset{(b)}{\le} \sum_{j=1}^{2m} (\tilde{\boldsymbol{w}}'_j - \hat{\boldsymbol{w}}_j) \tilde{f}(x_j) + 2\varepsilon \tag{119}$$

$$\le \sqrt{\frac{d}{2K} \log \frac{1}{\delta}} + \frac{\sqrt{d}}{3K} \log \frac{1}{\delta} + 2\varepsilon + \frac{8}{3\sqrt{d}}\varepsilon^2 + 2\varepsilon \tag{120}$$

$$\le \sqrt{\frac{d}{2K} \log \frac{1}{\delta}} + \frac{\sqrt{d}}{3K} \log \frac{1}{\delta} + 4\varepsilon + \frac{8}{3\sqrt{d}}\varepsilon^2 \tag{121}$$

$$\overset{(c)}{\le} \sqrt{\frac{d}{2K} \log \frac{1}{\delta}} + \frac{\sqrt{d}}{3K} \log \frac{1}{\delta} + 4\sqrt{d}\left(2^{\frac{d-2}{d+2}} \log^{\frac{1}{d+2}}(3)K^{-\frac{1}{d+2}} + \frac{1}{2}\sqrt{\frac{1}{K} \log \frac{1}{\delta}}\right) \tag{122}$$

$$+ \frac{8}{3}\sqrt{d}\left(2^{\frac{d-2}{d+2}} \log^{\frac{1}{d+2}}(3)K^{-\frac{1}{d+2}} + \frac{1}{2}\sqrt{\frac{1}{K} \log \frac{1}{\delta}}\right)^2 \tag{123}$$

$$= \sqrt{\frac{d}{2K} \log \frac{1}{\delta}} + \frac{\sqrt{d}}{3K} \log \frac{1}{\delta} + 4\sqrt{d}\left(2\left(\frac{\log 3}{2^4}\right)^{\frac{1}{d+2}} K^{-\frac{1}{d+2}} + \frac{1}{2}\sqrt{\frac{1}{K} \log \frac{1}{\delta}}\right) \tag{124}$$

$$+ \frac{8}{3}\sqrt{d}\left(2\left(\frac{\log 3}{2^4}\right)^{\frac{1}{d+2}} K^{-\frac{1}{d+2}} + \frac{1}{2}\sqrt{\frac{1}{K} \log \frac{1}{\delta}}\right)^2 \tag{125}$$

$$\le \sqrt{\frac{d}{2K} \log \frac{1}{\delta}} + \frac{\sqrt{d}}{3K} \log \frac{1}{\delta} + 4\sqrt{d}\left(2K^{-\frac{1}{d+2}} + \frac{1}{2}\sqrt{\frac{1}{K} \log \frac{1}{\delta}}\right) \tag{126}$$

$$+ \frac{8}{3}\sqrt{d}\left(2K^{-\frac{1}{d+2}} + \frac{1}{2}\sqrt{\frac{1}{K} \log \frac{1}{\delta}}\right)^2 \tag{127}$$

$$\le 3\sqrt{\frac{d}{K} \log \frac{1}{\delta}} + \frac{\sqrt{d}}{K} \log \frac{1}{\delta} + 8\sqrt{d}K^{-\frac{1}{d+2}} + 11\sqrt{d}K^{-\frac{2}{d+2}} + 6\sqrt{d \log \frac{1}{\delta}}K^{-\frac{1}{d+2}-\frac{1}{2}} \tag{128}$$

where (a) follows from $\|f - \tilde{f}\|_\infty \le \varepsilon$ and (b) follows from $\tilde{\boldsymbol{w}}', \hat{\boldsymbol{w}} \in \Delta_{2m-1}$ (c) follows from the definition of $\varepsilon$, and (d) follows from $d \ge 1$. From the Kantorovich-Rubinstein duality, we have

$$W_1(\mu_T^{\mathcal{D}_T}, \mu_T^{\hat{\boldsymbol{w}}}) \le 3\sqrt{\frac{d}{K} \log \frac{1}{\delta}} + \frac{\sqrt{d}}{K} \log \frac{1}{\delta} + 8\sqrt{d}K^{-\frac{1}{d+2}} + 11\sqrt{d}K^{-\frac{2}{d+2}} + 6\sqrt{d \log \frac{1}{\delta}}K^{-\frac{1}{d+2}-\frac{1}{2}}. \tag{129}$$

Note that we restricted test functions to $f(\frac{1}{2}) = 0$ but this does not affect the bound by adding a bias. As $\hat{\boldsymbol{w}}$ is the optimal solution of the linear program, we have $W_1(\mu_T^{\hat{\boldsymbol{w}}}, \mu_S) = \text{OPT}^{\text{continuous}} \le \text{OPT}^{\text{combinatorial}}$. By the triangle inequality, we have the desired guarantee. $\qquad\square$

