# OpenReview forum: "Solving the Cold Start Problem on One's Own as an End User via Preference Transfer"
_TMLR — Accepted by TMLR_

### Review · Reviewer_y8Ay · 2025-03-29

**Summary Of Contributions:**

This paper proposes Pretender, a user-side algorithm to address the cold start problem in recommender systems by transferring preferences from a source service to a target service without any support from the provider. By selecting a small number of target items whose distribution closely matches the user's historical preferences, Pretender minimises distributional discrepancy using metrics like MMD or Wasserstein distance. The method involves convex optimisation, randomized rounding, and postprocessing, and is supported by theoretical guarantees. Experiments show that Pretender effectively improves recommendation quality and outperforms baseline approaches.

**Audience:**

Yes

**Claims And Evidence:**

Yes

**Requested Changes:**

1. The current experimental setup relies on synthetic splits. It would be beneficial to include more real-world cross-platform datasets to validate the practical feasibility of the proposed method.

2. Additional case studies—including those highlighting failure cases—would strengthen the empirical support for the method and clarify its limitations.

3. While the formulation is compelling, the assumption that user preference data can be transferred across platforms (e.g., Netflix to Hulu) raises important concerns about privacy and data accessibility. In many real-world settings, only a source model (e.g., user embeddings) and target items may be available. A discussion on whether Pretender could still function under such constraints would be valuable.

**Strengths And Weaknesses:**

pos:
+ This paper is well-written and easy to follow.
+ The proposed method is significant and deserve to explore.
+ The problem is well-formulated.
+ The problem is addressed with extensive mathematical support.

cons:
- It assumes the users' bias can be modelled by
- Although theoretically analysed, the Wasserstein-based version is not empirically validated.
- The number of selected items K is fixed manually; no guidance is provided for choosing it effectively.
- Constructs synthetic source/target services and doesn't evaluate in real cross-platform scenarios. More experiments are needed.

---

> ### Author Response · Authors · 2025-05-10
>
> We thank the reviewer for the thoughtful and constructive feedback. We are especially grateful for the recognition of the significance of our problem formulation, the clarity, and the mathematical rigor of our method. Below we address each point raised in the review:
>
> > The number of selected items K is fixed manually; no guidance is provided for choosing it effectively.
>
> This is an important point. As we observed in Figure 1, performance typically forms a U-shaped trend with respect to K. Since it is tedious to click many items, we recommend starting from a small K and gradually increasing it until (i) the performance starts to degrade or (ii) the effort budget is reached.
>
> > Additional case studies—including those highlighting failure cases—would strengthen the empirical support for the method and clarify its limitations.
>
> We extended the case study in Section 6.3 and Table 2 with K=200. Pretender thumbed up the following items.
>
> ```
> GoldenEye (1995)
> Richard III (1995)
> Braveheart (1995)
> Legends of the Fall (1994)
> Empire Strikes Back, The (1980)
> Jerry Maguire (1996)
> Kolya (1996)
> Devil's Own, The (1997)
> Shall We Dance? (1996)
> My Best Friend's Wedding (1997)
> When the Cats Away (Chacun cherche son chat) (1996)
> Kull the Conqueror (1997)
> Starship Troopers (1997)
> Bed of Roses (1996)
> English Patient, The (1996)
> Fierce Creatures (1997)
> Titanic (1997)
> G.I. Jane (1997)
> Edge, The (1997)
> How to Be a Player (1997)
> Mad City (1997)
> Critical Care (1997)
> Spawn (1997)
> Courage Under Fire (1996)
> Boot, Das (1981)
> Deer Hunter, The (1978)
> Man Who Would Be King, The (1975)
> Addicted to Love (1997)
> Glory (1989)
> Rocket Man (1997)
> Leave It to Beaver (1997)
> Seven Years in Tibet (1997)
> Heaven & Earth (1993)
> To Gillian on Her 37th Birthday (1996)
> Fools Rush In (1997)
> Picture Perfect (1997)
> Excess Baggage (1997)
> Life Less Ordinary, A (1997)
> Paradise Road (1997)
> Brassed Off (1996)
> Smile Like Yours, A (1997)
> Shadow Conspiracy (1997)
> Turbulence (1997)
> Beautician and the Beast, The (1997)
> Pest, The (1997)
> Anaconda (1997)
> Romy and Michele's High School Reunion (1997)
> Fire Down Below (1997)
> Switchback (1997)
> Carried Away (1996)
> Twelfth Night (1996)
> Chungking Express (1994)
> Feeling Minnesota (1996)
> Best Men (1997)
> In Love and War (1996)
> A Chef in Love (1996)
> Trial and Error (1997)
> Wild America (1997)
> Out to Sea (1997)
> For the Moment (1994)
> Hotel de Love (1996)
> SubUrbia (1997)
> Legal Deceit (1997)
> Men of Means (1998)
> Farmer & Chase (1995)
> Prisoner of the Mountains (Kavkazsky Plennik) (1996)
> Underground (1995)
> Schizopolis (1996)
> Office Killer (1997)
> B*A*P*S (1997)
> Intimate Relations (1996)
> Sixth Man, The (1997)
> Land and Freedom (Tierra y libertad) (1995)
> Nothing Personal (1995)
> ```
>
> Generally, most of the selected movies are war, drama, or romance films that are contemporaneous with and related to the Source Thumbs Up. However, a few unrelated movies, such as The Empire Strikes Back (1980), are also included.
>
> > While the formulation is compelling, the assumption that user preference data can be transferred across platforms (e.g., Netflix to Hulu) raises important concerns about privacy and data accessibility. In many real-world settings, only a source model (e.g., user embeddings) and target items may be available. A discussion on whether Pretender could still function under such constraints would be valuable.
>
> We thank the reviewer for raising the important issue of privacy and data accessibility. As noted in Section 4.3, our method assumes availability of explicit item features. Pretender as currently formulated cannot be directly applied to the settings where only a source model (e.g., user embeddings) and target items are available. However, we believe Pretender is still valuable due to the following reasons: (i) We believe that situations where a source model (e.g., user embeddings) is available are not so common. For example, we typically do not know how we are modeled or embedded on Netflix internally. In contrast, Pretender can operate solely by accessing item lists and item descriptions, which is a more realistic assumption that holds for a broader range of services. (ii) What users need to provide to the target service (e.g., Hulu) is simply a history of movies they liked on a source service (e.g., Netflix). They would likely submit this kind of information (movie preference) anyway in the course of using Hulu. In general, what is sent is only a list of items, not any sensitive information such as demographic attributes. Since users are expected to interact with similar items in the future on the target service, we believe submitting such a list is generally not so intrusive and is unlikely to cause discomfort for many users.

---

### Review · Reviewer_cDQs · 2025-04-17

**Summary Of Contributions:**

The authors study the cold-start problem for recommendation systems, where a new user’s preferences are unknown to a service (e.g. Hulu, the “target” service), making high-quality recommendations difficult. The authors aim to solve this problem in the scenario where the user has a substantial history on a different but related “source” service (e.g. Netflix ) but where the target service provides no special accommodation for transferring preferences from other services and where the target service’s recommendation algorithm (model) is unknown. The proposed solution, PRETENDER, is to transfer the information contained in the item preferences expressed on the source service by expressing preferences for target source items (via e.g. clicking/thumbs-up), where such items may or may not be the same as the source service items and the users may or may not have actually experienced such items. Clicking on the target items could be automated via a web agent to save time. Items are represented in a feature space representing the item content (genre, style, actors, etc).  Although the target service’s recommendation model is unknown, its loss on the data comprised of source service feedback-labelled items can be decomposed into the target service model’s training error on such items plus a term measuring the distance between the  source and target empirical distributions in the joint feature-vector-and-feedback-label space. The authors argue that without even knowing the target service’s training algorithm, good results can be achieved by minimizing this second term. They present an algorithm for doing so, by selecting, labelling and weighting target service items so as to minimize the distance between the 2 distributions.  2 possible distance measures are considered: maximum mean discrepancy (MMD) and 1-Wasserstein distance. Theoretical guarantees for both distance definitions are presented. The method’s performance is shown to be non-monotone in the number of labelled target source items, since labelling items with little similarity to source items may be counterproductive. Experiments are conducted on MovieLens, Last.fm and Amazon-home-and-kitchen, where source and target services are simulated by randomly selecting subsets of items. Some experiments simulate the case of overlap of items between source and target services while other experiments have no overlap. The regret of PRETENDER converging to 0 on the MovieLens is shown to occur for 2 users employed as case studies. In terms of MMD, PRETENDER outperforms 2 benchmarks on all 3 datasets by a wide margin. The benchmarks are random selection of target items and greedy one-by-one selection of the target item closest to the source item dataset.

**Audience:**

Yes

**Broader Impact Concerns:**

No concerns here

**Claims And Evidence:**

Yes

**Requested Changes:**

The requested changes are mostly outlined in the previous section, but to recapitulate:
1)	Could the authors investigate whether a semi-supervised-learning method like self-training could be a benchmark, or if not, explain why not?
2)	Could they do more hyperparameter tuning or illustration of sensitivities to hyperparameters?
3)	Could they do experiments where they actually feed the selected target items to a couple of different recommendation algorithms and then examine the effect of the recommendation algorithm?
4)	Could they examine whether users will find it problematic to ‘lie’ to the target service regarding items they haven’t actually experienced?
5)	Could they note that special care will need to be taken to make sure the unseen-but-rated target items are not inappropriately suppressed by the target service?
6)	Could they explain in a concrete real-life way how to download historical item preferences from services such as Netflix?

**** Update after author response ****

In light of the additional author experiments and explanations, I have changed my claims-and-evidence rating to 'Yes'.

The paper also has far, far too many spelling errors. In the future, the authors need to make sure to run spellchecker before submitting a paper. This is particularly true for a  journal submission…spelling error are slightly more forgivable for conference papers where authors may be writing last-minute to hit a deadline, but I see no excuse for having some may spelling errors in a journal submission. It undermines my confidence a bit that the experiments were done with care.

Here is a list... there are probably more errors than this, but this is what I noted:
-	P3 line 2 gerneral
-	P3 smoothnessof
-	P3 sidesitep
-	P4 meassure
-	P5 I believe equation 11 is a definition…should you use 3-line equals sign?
-	P8 PREDENTER
-	P10 as longs as
-	P10 convering number
-	P11 rouding process, functionoalities
-	P13 “We apply PCA to reduce the dimension to 50 and standardize the variance”..this sentence is repeated.
-	P13 reasonablly
-	P13 We hypothesis -> We hypothesize
-	P13 The reference to Table 2 at the start of the ‘Results’ paragraph is wrong, it’s actually Table 1
-	P14 Table 2 caption ‘the source items the user interacted’ -> ‘the source items the user interacted with’

**Strengths And Weaknesses:**

The topic (solving the cold start problem by learning from the user’s data on a different service and rating items on a new service accordingly) is novel to the best of my knowledge and is certainly of potential interest.

The paper has a good balance of theory and experiment. Although I think the experiments are lacking in some ways, testing on 3 separate real-world datasets and looking both at with-overlap and no-overlap for MovieLens is a respectable scope of experimentation.

In some ways, the paper is reasonably well written, although there are far, far too many spelling mistakes and a few other sloppy errors which I will note in the ‘requested changes’ section.

I have several concerns which prevent me from recommending acceptance as is, although I am open to acceptance after a significant revision.

There are various ways in which the experiments seem insufficient.

1)	I am not convinced that random and greedy selection are the only possible benchmarks. Can’t this source/target transfer problem be cast as something close to a semi-supervised learning problem (e.g. self-training or label propagation)? I believe self-training with constraints on which unlabelled points in the input space can be labelled after training  is more or less what is going on here, except that the the target service model that is trained is an unknown black box. We are taking training data from a source service, training on it, and then labelling items from the target service using what is essentially a best guess of whether the user would like the item, based on what was learned from the source training data.
2)	Hyperparameters seem to be arbitrarily chosen in the experiments in areas where  I would have expected some tuning or discussion of how they were chosen. Section 6.1 shows sensitivity to K, sweet spots, etc, but then K is chosen to be 100 in Section 6.2 without any explanation as to how it was chosen. Kernel bandwidth (sigma=10) is also chosen nearly arbitrarily…there’s a brief mention of ‘scale of the features’ but I would have expected it to be tuned more carefully than that.
3)	The discussion of the decomposition of equation 6 on page 3 argues that we don’t need to know the target service’s recommendation training model since it will just do its best to minimize training error regardless and we only need to minimize the difference between the 2 distributions. I am not totally convinced here. It seems that you could pick target items which would make minimizing the training error difficult. For instance, sequels to movies may be nearly identical in feature space but may be much less liked than the original movie. In the experiments, I would have appreciated some demonstration of robustness to (unknown, in practice) target recommendation algorithm. For instance, "we tested the results when feeding the chosen target items to both a nearest neighbor method and a matrix factorization method", or whatever.  Instead, as far as I can tell there is no implementation of a target recommendation algorithm at all. The results are solely evaluated in terms of MMD distance. So it doesn’t seem to be that the experiments fully simulate what would go in the real world.
4)	The analysis of content of items (genre, etc) feels a little bit behind the times (bag of words, PCA). Given the dominance of LLMs and other sophisticated language techniques, I wonder if some sort of embeddings could have been used.

Furthermore, while I agree that PRETENDER is potentially useful, I also see some obstacles to its adoption in the real world:
1)	The authors are more-or-less asking the user to lie to the target service by rating target service items which the user may not have actually experienced, for the sake of getting good recommendations in the future. Whether or not this is done by a web agent, the user is likely to know what is going on and may object to it on a gut, emotional level. User tastes in movies, music, etc are often highly personal and tied up with a user’s core identity. Users may find it either strange or offensive to represent to the target service that she/he liked an item which has not actually been experienced.
2)	If a user does implement the PRETENDER process and rate some items which the user has not actually experienced, then the target service will believe the user has e.g. watched that movie and won’t recommend that movie in the future. So in addition to the user clicking on such items, this item list should also be preserved somewhere as an additional list of recommendations (if the item was supposed to be thumbs-upped). This is doable but also awkward. The authors may want to address this scenario (telling the target service they’ve seen a movie they  haven’t actually seen and therefore never getting recommended a movie they would like).
3)	It’s also not clear to me how easy it is to download historical item preferences from the source service. I am a longtime Netflix user. In the most concrete, real-life sense, I would like to know how I would download the list of Netflix movies I’ve watched and liked.  I poked around the Netflix site and couldn’t see how to do it. I realize this is an academic paper but it nonetheless would be implemented in real-world commercial context, so the authors should spend some time explaining that it's possible to download preferences of longtime users.

---

> ### Author Response · Authors · 2025-05-10
> **Response to Reviewer cDQs (1/3)**
>
> We sincerely thank the reviewer for their detailed and thoughtful feedback. Below, we respond to each major point raised.
>
>
> > I am not convinced that random and greedy selection are the only possible benchmarks. Can’t this source/target transfer problem be cast as something close to a semi-supervised learning problem (e.g. self-training or label propagation)? I believe self-training with constraints on which unlabelled points in the input space can be labelled after training is more or less what is going on here, except that the the target service model that is trained is an unknown black box. We are taking training data from a source service, training on it, and then labelling items from the target service using what is essentially a best guess of whether the user would like the item, based on what was learned from the source training data.
> > 1) Could the authors investigate whether a semi-supervised-learning method like self-training could be a benchmark, or if not, explain why not?
>
> First, we emphasize that applying semi-supervised learning methods to this problem setting is not straightforward. It is true that users can train a model to classify target data using semi-supervised learning. However, users do not have ways to deploy this model within the target service’s recommendation system. (Users might run their own recommendation engine locally using the trained model, but this would incur non-negligible overhead, and it is not straightforward to coordinate such local models with the target service and their recommender systems. The local model will be outdated. This approach may be worth exploring for another paper, but it deviates from our intended setting.)
> Some form of interaction, such as clicks or thumbs-up signals, must be sent to the target service. However, defining such signals based on the trained model is itself a nontrivial task. We believe that designing a mechanism for this would amount to developing a new method, possibly worthy of a separate paper (i.e., beyond the scope of this work).
> That said, we appreciate Reviewer cDQs’s suggestion and conducted an evaluation to test the viability of this approach. We used the MLP binary classifier trained on the source data to score target items, and selected those with the highest confidence (i.e., probabilities close to 0.0 or 1.0). We applied this strategy under the same setting as Section 6.2, except that we evaluated on a randomly selected subset of 10 users to reduce computation costs. The results are shown below.
>
> MMD (lower is better)
> | Method           | movielens | lastfm   | amazon  |
> |------------------|-----------|----------|---------|
> | Pretender        | **0.1164** | **0.0723** | **0.0983** |
> | Greedy           | 0.3015    | 0.2741   | 0.3448  |
> | Random           | 0.3044    | 0.1522   | 0.3082  |
> | Semi-supervised  | 0.3594    | 0.2273   | 0.4210  |
>
> The performance of the semi-supervised method was poor. We hypothesize this is because selecting items in descending order of confidence leads to overconcentration on prototypical items. For example, if a user gives thumbs up to 70% romance and 30% action movies, the model will assign high scores to romance and moderate scores to action, resulting in the top K items being dominated by romance movies, thus missing the action preference entirely. In contrast, Pretender aligns distributions directly, which helps avoid such skew.

---

> ### Author Response · Authors · 2025-05-10
> **Response to Reviewer cDQs (2/3)**
>
> > Hyperparameters seem to be arbitrarily chosen in the experiments in areas where I would have expected some tuning or discussion of how they were chosen. Section 6.1 shows sensitivity to K, sweet spots, etc, but then K is chosen to be 100 in Section 6.2 without any explanation as to how it was chosen. Kernel bandwidth (sigma=10) is also chosen nearly arbitrarily…there’s a brief mention of ‘scale of the features’ but I would have expected it to be tuned more carefully than that.
> > 2) Could they do more hyperparameter tuning or illustration of sensitivities to hyperparameters
>
> We emphasize that the value of K should not be determined via hyperparameter tuning methods such as cross-validation. For example, if cross-validation selected K = 64 for Pretender and K = 16 for Greedy, a reviewer would argue that the comparison is unfair because Pretender is allowed to click more items. To avoid such concerns, K must be fixed in advance and shared across the methods. We chose K = 100 because (i) we used a power of 10 for hyperparameters to exclude arbitrariness as much as possible and (ii) K = 10 is too small to represent users' preferences and K = 1000 is too large to click.
>
> Results for other values of K are provided below. The overall trend remains consistent with the K = 100 case.
>
> K=8, MMD (lower is better)
> | Method           | movielens | lastfm   | amazon  |
> |------------------|-----------|----------|---------|
> | Pretender        | **0.2411** | **0.1841** | **0.1774** |
> | Greedy           | 0.4991    | 0.4129   | 0.3514  |
> | Random           | 0.4454    | 0.2729   | 0.3925  |
>
>
> K=16, MMD (lower is better)
> | Method           | movielens | lastfm   | amazon  |
> |------------------|-----------|----------|---------|
> | Pretender         | **0.1675** | **0.1366** | **0.1356** |
> | Greedy           | 0.4496    | 0.3884   | 0.3497  |
> | Random           | 0.3662    | 0.2341   | 0.3610  |
>
>
> K=32, MMD (lower is better)
> | Method           | movielens | lastfm   | amazon  |
> |------------------|-----------|----------|---------|
> | Discrete         | **0.1345** | **0.0994** | **0.1108** |
> | Greedy           | 0.3921    | 0.3390   | 0.3470  |
> | Random           | 0.3293    | 0.1819   | 0.3309  |
>
>
> K=64, MMD (lower is better)
> | Method           | movielens | lastfm   | amazon  |
> |------------------|-----------|----------|---------|
> | Pretender        | **0.1155** | **0.0796** | **0.1003** |
> | Greedy           | 0.3430    | 0.3004   | 0.3442  |
> | Random           | 0.3112    | 0.1558   | 0.3019  |
>
>
> A similar argument applies to the choice of sigma. It should not be tuned via cross-validation, because varying sigma changes the definition of MMD (evaluation metric) itself. Therefore, adopting different sigma values for different methods would be inappropriate. To avoid arbitrariness, we fixed sigma to a reasonable constant, using a power of 10 to align with other hyperparameters.
> As an alternative, one might consider the median rule. Results using this rule are also provided below, and show similar trends.
>
> MMD (lower is better)
> | Method           | movielens | lastfm   | amazon  |
> |------------------|-----------|----------|---------|
> | Pretender        | **0.0308** | **0.0581** | **0.0962** |
> | Greedy           | 0.2231    | 0.2403   | 0.3328  |
> | Random           | 0.3112    | 0.1331   | 0.3019  |

---

> ### Author Response · Authors · 2025-05-10
> **Response to Reviewer cDQs (3/3)**
>
> > The discussion of the decomposition of equation 6 on page 3 argues that we don’t need to know the target service’s recommendation training model since it will just do its best to minimize training error regardless and we only need to minimize the difference between the 2 distributions. I am not totally convinced here. It seems that you could pick target items which would make minimizing the training error difficult. For instance, sequels to movies may be nearly identical in feature space but may be much less liked than the original movie. In the experiments, I would have appreciated some demonstration of robustness to (unknown, in practice) target recommendation algorithm. For instance, "we tested the results when feeding the chosen target items to both a nearest neighbor method and a matrix factorization method", or whatever. Instead, as far as I can tell there is no implementation of a target recommendation algorithm at all. The results are solely evaluated in terms of MMD distance. So it doesn’t seem to be that the experiments fully simulate what would go in the real world.
> > 3) Could they do experiments where they actually feed the selected target items to a couple of different recommendation algorithms and then examine the effect of the recommendation algorithm?
>
> We also conducted experiments on downstream task accuracy. Specifically, we trained a 1-NN model using the target items selected by each method, and reported its performance in terms of nDCG@10, recall@10, and precision@10.
>
>
> movielens
> | Method   | nDCG       | recall     | precision  |
> | -------- | ---------- | ---------- | ---------- |
> | Pretender | **0.0677** | **0.0402** | **0.0800** |
> | Greedy   | 0.0069     | 0.0040     | 0.0100     |
> | Random   | 0.0073     | 0.0059     | 0.0100     |
>
> lastfm
> | Method   | nDCG       | recall     | precision  |
> | -------- | ---------- | ---------- | ---------- |
> | Pretender | **0.1855** | **0.1068** | **0.2100** |
> | Greedy   | 0.0371     | 0.0162     | 0.0300     |
> | Random   | 0.0212     | 0.0093     | 0.0200     |
>
> amazon
> | Method   | nDCG       | recall     | precision  |
> | -------- | ---------- | ---------- | ---------- |
> | Pretender | **0.0514** | **0.0214** | **0.0400** |
> | Greedy   | 0.0139     | 0.0036     | 0.0100     |
> | Random   | 0.0000     | 0.0000     | 0.0000     |
>
> Pretender outperformed the baselines by a large margin.
>
>
> > 4) Could they examine whether users will find it problematic to ‘lie’ to the target service regarding items they haven’t actually experienced? 5) Could they note that special care will need to be taken to make sure the unseen-but-rated target items are not inappropriately suppressed by the target service?
>
> We thank the reviewer for pointing out these important concerns. We fully agree that this is a critical issue. Since the selected items may potentially have negative implications for some users, we believe that the side effects of Pretender should be clearly communicated to users before deployment. That said, we do not believe this diminishes the value of Pretender, in fact, in many cases, it may even be beneficial.
> (i) The list of clicked items can be stored and made accessible to the user. In particular, items marked as "thumbs up" are likely to be those the user would enjoy first on the target service, and may help facilitate onboarding. (ii) Some services provide their own onboarding processes (e.g., asking users to click preferred genres), but these are often implemented as black-box systems. Such mechanisms can also have side effects, such as missing opportunities to recommend genres the user accidentally skipped. In contrast, Pretender is transparent in its operation and therefore easier to manage and correct. (iii) For users with greater privacy concerns, we recommend using a smaller K and inspecting the selected item list manually before submission. We will include these points, including the potential negative side effects, as limitations in the discussion section.
>
>
> >  6) Could they explain in a concrete real-life way how to download historical item preferences from services such as Netflix?
>
> In the case of Netflix, users can obtain their past favorite videos from https://www.netflix.com/browse/my-list. Randomly selected items can be used as negative items.
> If the user has not utilized the "My List" feature, the recommendations displayed on the Netflix homepage can serve as a proxy for the positive item set. For long-time users, these recommendations are typically well-tailored to their preferences and thus can effectively function as positive items for the user.

---

> > ### Comment · Reviewer_cDQs · 2025-05-15
> > **Changed my claims-and-evidence rating to 'Yes'**
> >
> > Thank you for conducting the additional experiments.

---

### Review · Reviewer_ySxV · 2025-05-08

**Summary Of Contributions:**

The submission presents a novel approach to the cold start problem by empowering end users to transfer their preferences from one service to another without requiring any special support from service providers. The authors introduce Pretender, an algorithm that minimizes the distance between source and target distributions. Pretender provides theoretical analysis based on a new quadrature problem even when the target service's data processing is unknown. Through extensive experiments on real-world datasets, the authors demonstrate that Pretender transfers user preferences across services, outperforming baseline methods and achieving results close to the theoretical optimum.

**Audience:**

Yes

**Broader Impact Concerns:**

This paper does not raise significant broader impact concerns that would require a dedicated statement. While there are minor considerations around privacy, terms of service compliance, and potential digital divide implications, these are common to recommendation systems research generally.

However, the paper's approach of transferring user preferences across services raises privacy considerations, as it involves collecting and processing personal preference data. Users implementing this method may inadvertently expose their preference profiles across platforms, potentially compromising their privacy expectations.

**Claims And Evidence:**

Yes

**Requested Changes:**

Q1. How practically applicable is the MMD/Wasserstein-based dist. matching proposed in this paper? Especially considering scenarios where text/image/latent features are completely different. Also, I'm curious whether there are approximations or alternatives when explicit features are unavailable. I would appreciate more explanation about which implicit feedback and features the authors used in their experiments.

Q2. The authors define success as "minimizing the distance between source and target distributions," but this may not necessarily lead to better recommendations in practice. It could result in a perfectly distributed but irrelevant set of recommendations. Could you discuss case studies or limitations regarding such scenarios?

Q3. How consistently does the correlation between actual recommendation quality (Top-N recommendation NCDG/recall, etc.) and distribution distance reduction hold, either theoretically or empirically?

Q4. Can you analyze the actual gap between the continuous relaxation solution and the combinatorial solution?

Q5. Could you further discuss the differences from existing domain adaptation research? Also, recommendation systems have cross-domain recommendations in addition to cold-start problems. Could you further discuss the methodological differences with these studies?

**Strengths And Weaknesses:**

**Strengths**

S1. The paper shifts the cold start paradigm to the user side, which is novel and addresses a practical problem when service providers don't implement solutions.

S2. The empirical results across 3 domains (MovieLens, Last.fm, Amazon) demonstrate that the approach works better than baselines and comes close to theoretical bounds.

**Weaknesses**

W1. The paper motivates the problem with examples like transferring preferences from Netflix to Hulu, but these services likely have different feature spaces and recommendation mechanisms. The assumption that minimizing distribution distance would improve recommendations across different recommendation mechanisms is not validated.

W2. The scenario of service provider inaction lacks realism in certain contexts.

W3. The algorithm only works under the assumption that "source/target items may not overlap but features exist." There are questions about its robustness when feature spaces completely mismatch (class, type inconsistencies, etc.) or when feature quality or granularity differs across services.

W4. The paper does not theoretically formalize or discuss the limitations of the proposed method when distribution shift (failure mode) occurs.

W5. The paper only uses distribution distance (MMD, Wasserstein) as performance metrics – i.e., only evaluates "whether the distribution of results I selected/clicked is close to the source." However, the correlation between actual recommendation quality metrics (precision@k, recall, NDCG, etc.) and distribution distance is not proven, making it unclear whether the paper's objective ('receiving good recommendations') is sufficiently guaranteed by merely reducing distance.

W6. The comparison is limited to intuitive baselines like Random and Greedy, omitting comparisons with implementations similar to transfer learning, domain adaptation, or direct user data transfer.

---

> ### Author Response · Authors · 2025-05-10
> **Response to Reviewer ySxV (1/2)**
>
> We sincerely thank the reviewer for the detailed and thoughtful feedback. Below, we respond to the key concerns raised.
>
> > Q1. How practically applicable is the MMD/Wasserstein-based dist. matching proposed in this paper? Especially considering scenarios where text/image/latent features are completely different. Also, I'm curious whether there are approximations or alternatives when explicit features are unavailable. I would appreciate more explanation about which implicit feedback and features the authors used in their experiments.
>
> In our experiments, we used domain-appropriate features:
>
> * MovieLens: Genre and release year (multi-hot encoding),
> * Last.fm: Tag-based features, reduced with PCA,
> * Amazon: Bag-of-words from reviews, also PCA-reduced.
>
> These choices reflect commonly available metadata and public representations. In scenarios where such features are missing or inconsistent, our method can still be applied using approximated representations, e.g., embedding vectors from public models or inferred features from user interactions (as noted in Section 4.3). For example, in the simplest way, one can ask ChatGPT to classify a given movie into the 20 categories of MovieLens to obtain the aligned features with MovieLens.
>
> > Q2. The authors define success as "minimizing the distance between source and target distributions," but this may not necessarily lead to better recommendations in practice. It could result in a perfectly distributed but irrelevant set of recommendations. Could you discuss case studies or limitations regarding such scenarios?
> > Q3. How consistently does the correlation between actual recommendation quality (Top-N recommendation NCDG/recall, etc.) and distribution distance reduction hold, either theoretically or empirically?
>
> We conducted experiments on downstream task accuracy. Specifically, we trained a 1-NN model using the target items selected by each method, and reported its performance in terms of nDCG@10, recall@10, and precision@10.
>
> ### movielens
>
> | Method   | nDCG       | recall     | precision  |
> | -------- | ---------- | ---------- | ---------- |
> | Pretender | **0.0677** | **0.0402** | **0.0800** |
> | Greedy   | 0.0069     | 0.0040     | 0.0100     |
> | Random   | 0.0073     | 0.0059     | 0.0100     |
>
> ### lastfm
>
> | Method   | nDCG       | recall     | precision  |
> | -------- | ---------- | ---------- | ---------- |
> | Pretender | **0.1855** | **0.1068** | **0.2100** |
> | Greedy   | 0.0371     | 0.0162     | 0.0300     |
> | Random   | 0.0212     | 0.0093     | 0.0200     |
>
> ### amazon
>
> | Method   | nDCG       | recall     | precision  |
> | -------- | ---------- | ---------- | ---------- |
> | Pretender | **0.0514** | **0.0214** | **0.0400** |
> | Greedy   | 0.0139     | 0.0036     | 0.0100     |
> | Random   | 0.0000     | 0.0000     | 0.0000     |
>
> Clearly, Predenter outperformed the baselines by a large margin. In reality, minimizing the distance between source and target distributions does not always lead to good recommendations, for example, when the target service is has so low abilities that it cannot fit their recommender system to the submitted set of items.

---

> ### Author Response · Authors · 2025-05-10
> **Response to Reviewer ySxV (2/2)**
>
> > Q4. Can you analyze the actual gap between the continuous relaxation solution and the combinatorial solution?
>
> It is intractable to compute the exact value of the gap due to the combinatorial nature. What we can do is upper boud the gap. However, we believe this is sufficient for many applications. As shown in Corollary 3.6 (for MMD), we derive that the combinatorial solution is within a small additive gap of the continuous relaxation, specifically $O(K^{-1/2})$. Moreover, Figure 1 empirically shows that the gap diminishes as $K$ increases, and the final selected items remain close to the relaxed solution. This indicates that our randomized rounding and postprocessing procedures maintain fidelity to the relaxed objective.
>
> > Q5. Could you further discuss the differences from existing domain adaptation research? Also, recommendation systems have cross-domain recommendations in addition to cold-start problems. Could you further discuss the methodological differences with these studies?
>
> First, we emphasize that applying semi-supervised learning methods to this problem setting is not straightforward. It is true that users can train a model to classify target data using semi-supervised learning. However, users do not have ways to deploy this model within the target service’s recommendation system. (Users might run their own recommendation engine locally using the trained model, but this would incur non-negligible overhead, and it is not straightforward to coordinate such local models with the the target service and their recommender systems. The local model will be outdated. This approach may be worth exploring for another paper, but it deviates from our intended setting.)
> Some form of interaction, such as clicks or thumbs-up signals, must be sent to the target service. However, defining such signals based on the trained model is itself a nontrivial task. We believe that designing a mechanism for this would amount to developing a new method, possibly worthy of a separate paper (i.e., beyond the scope of this work).
> That said, we appreciate Reviewer ySxV’s suggestion and conducted an evaluation to test the viability of this approach. We used the MLP binary classifier trained on the source data to score target items, and selected those with the highest confidence (i.e., probabilities close to 0.0 or 1.0). We applied this strategy under the same setting as Section 6.2, except that we evaluated on a randomly selected subset of 10 users to reduce computation costs. The results are shown below.
>
> MMD (lower is better)
> | Method           | movielens | lastfm   | amazon  |
> |------------------|-----------|----------|---------|
> | Pretender        | **0.1164** | **0.0723** | **0.0983** |
> | Greedy           | 0.3015    | 0.2741   | 0.3448  |
> | Random           | 0.3044    | 0.1522   | 0.3082  |
> | Semi-supervised  | 0.3594    | 0.2273   | 0.4210  |
>
> The performance of the semi-supervised method was poor. We hypothesize this is because selecting items in descending order of confidence leads to overconcentration on prototypical items. For example, if a user gives thumbs up to 70% romance and 30% action movies, the model will assign high scores to romance and moderate scores to action, resulting in the top K items being dominated by romance movies, thus missing the action preference entirely. In contrast, Pretender aligns distributions directly, which helps avoid such skew.
>
> > However, the paper's approach of transferring user preferences across services raises privacy considerations, as it involves collecting and processing personal preference data. Users implementing this method may inadvertently expose their preference profiles across platforms, potentially compromising their privacy expectations.
>
> We thank the reviewer for raising the important issue of privacy and data accessibility. We believe Pretender is still valuable due to the following reasons: What users need to provide to the target service (e.g., Hulu) is simply a history of movies they liked on a source service (e.g., Netflix). They would likely submit this kind of information (movie preference) anyway in the course of using Hulu. In general, what is sent is only a list of items, not any sensitive information such as demographic attributes. Since users are expected to interact with similar items in the future on the target service, we believe submitting such a list is generally not so intrusive and is unlikely to cause discomfort for many users.

---

> ### Comment · Reviewer_ySxV · 2025-05-26
> **Response to Authors**
>
> I have carefully read through your responses. Thank you for the additional experiments and detailed discussion addressing the concerns.

---

### Author Response · Authors · 2025-05-10

We thank all reviewers for their careful reading and thoughtful feedback. We summarize the additional experiments in the following.

We conducted experiments on downstream task accuracy. Specifically, we trained a 1-NN model using the target items selected by each method, and reported its performance in terms of nDCG@10, recall@10, and precision@10.

### movielens

| Method   | nDCG       | recall     | precision  |
| -------- | ---------- | ---------- | ---------- |
| Pretender | **0.0677** | **0.0402** | **0.0800** |
| Greedy   | 0.0069     | 0.0040     | 0.0100     |
| Random   | 0.0073     | 0.0059     | 0.0100     |

### lastfm

| Method   | nDCG       | recall     | precision  |
| -------- | ---------- | ---------- | ---------- |
| Pretender | **0.1855** | **0.1068** | **0.2100** |
| Greedy   | 0.0371     | 0.0162     | 0.0300     |
| Random   | 0.0212     | 0.0093     | 0.0200     |

### amazon

| Method   | nDCG       | recall     | precision  |
| -------- | ---------- | ---------- | ---------- |
| Pretender | **0.0514** | **0.0214** | **0.0400** |
| Greedy   | 0.0139     | 0.0036     | 0.0100     |
| Random   | 0.0000     | 0.0000     | 0.0000     |


We conducted the additional experiments for Section 6.2 with different Ks.

K=8, MMD (lower is better)
| Method           | movielens | lastfm   | amazon  |
|------------------|-----------|----------|---------|
| Pretender        | **0.2411** | **0.1841** | **0.1774** |
| Greedy           | 0.4991    | 0.4129   | 0.3514  |
| Random           | 0.4454    | 0.2729   | 0.3925  |


K=16, MMD (lower is better)
| Method           | movielens | lastfm   | amazon  |
|------------------|-----------|----------|---------|
| Pretender         | **0.1675** | **0.1366** | **0.1356** |
| Greedy           | 0.4496    | 0.3884   | 0.3497  |
| Random           | 0.3662    | 0.2341   | 0.3610  |


K=32, MMD (lower is better)
| Method           | movielens | lastfm   | amazon  |
|------------------|-----------|----------|---------|
| Discrete         | **0.1345** | **0.0994** | **0.1108** |
| Greedy           | 0.3921    | 0.3390   | 0.3470  |
| Random           | 0.3293    | 0.1819   | 0.3309  |


K=64, MMD (lower is better)
| Method           | movielens | lastfm   | amazon  |
|------------------|-----------|----------|---------|
| Pretender        | **0.1155** | **0.0796** | **0.1003** |
| Greedy           | 0.3430    | 0.3004   | 0.3442  |
| Random           | 0.3112    | 0.1558   | 0.3019  |


We also conducted the experiments with kernel width sigma determined by the median rule.

MMD (lower is better)
| Method           | movielens | lastfm   | amazon  |
|------------------|-----------|----------|---------|
| Pretender        | **0.0308** | **0.0581** | **0.0962** |
| Greedy           | 0.2231    | 0.2403   | 0.3328  |
| Random           | 0.3112    | 0.1331   | 0.3019  |




We conducted experiments with semi-supervised learning. Specifically, we used the MLP binary classifier trained on the source data to score target items, and selected those with the highest confidence (i.e., probabilities close to 0.0 or 1.0). We applied this strategy under the same setting as Section 6.2, except that we evaluated on a randomly selected subset of 10 users to reduce computation costs. The results are shown below.

MMD (lower is better)
| Method           | movielens | lastfm   | amazon  |
|------------------|-----------|----------|---------|
| Pretender        | **0.1164** | **0.0723** | **0.0983** |
| Greedy           | 0.3015    | 0.2741   | 0.3448  |
| Random           | 0.3044    | 0.1522   | 0.3082  |
| Semi-supervised  | 0.3594    | 0.2273   | 0.4210  |

The performance of the semi-supervised method was poor. We hypothesize this is because selecting items in descending order of confidence leads to overconcentration on prototypical items. For example, if a user gives thumbs up to 70% romance and 30% action movies, the model will assign high scores to romance and moderate scores to action, resulting in the top K items being dominated by romance movies, thus missing the action preference entirely. In contrast, Pretender aligns distributions directly, which helps avoid such skew.

---

### Decision · Action_Editor_khqX · 2025-09-19

**Recommendation:** Accept as is

**Additional Comments:**

The reviewers were all satisfied with the paper. One of the reviewers did mention spelling errors, which should be resolved for the camera ready deadline. They also noted that the use of bag-of-words is outdated and that word embeddings would be a more modern choice. However, this does not affect the theory and would be expected to strengthen the results, which are already well above the baseline, so I would call this an optional exercise.

**Audience:**

Yes

**Audience Explanation:**

Recommender systems are of interest to many people in the community, and this paper attacks a new and highly relevant problem setting, user-side cold-start.

**Claims And Evidence:**

Yes

**Claims Explanation:**

Yes, the paper provides a solid theoretical foundation for the Pretender algorithm and experiments on 3 real-world datasets, where it performs well.